# Pan-genome analysis of six *Paracoccus* type strain genomes reveal lifestyle traits

**Jacqueline Hollensteiner**[1], **Dominik Schneider**[1], **Anja Poehlein**[1],
**Thorsten Brinkhoff**[2], **Rolf Daniel**[1]*

**1** Genomic and Applied Microbiology and Göttingen Genomics Laboratory, Institute of Microbiology and Genetics, Georg-August University of Göttingen, Göttingen, Germany, **2** Institute for Chemistry and Biology of the Marine Environment, University of Oldenburg, Oldenburg, Germany

* rdaniel@gwdg.de

**Data Availability Statement:** The whole-genome sequence project has been deposited at DDBJ/ENA/GenBank. Data are available for: P. aesturarii DSM 19484T under BioProject PRJNA689385, BioSample SAMN17208934, accession numbers

## Abstract

The genus *Paracoccus* capable of inhabiting a variety of different ecological niches both, marine and terrestrial, is globally distributed. In addition, *Paracoccus* is taxonomically, metabolically and regarding lifestyle highly diverse. Until now, little is known on how *Paracoccus* can adapt to such a range of different ecological niches and lifestyles. In the present study, the genus *Paracoccus* was phylogenomically analyzed (n = 160) and revisited, allowing species level classification of 16 so far unclassified *Paracoccus* sp. strains and detection of five misclassifications. Moreover, we performed pan-genome analysis of *Paracoccus*-type strains, isolated from a variety of ecological niches, including different soils, tidal flat sediment, host association such as the bluespotted cornetfish, *Bugula plumosa*, and the reef-building coral *Stylophora pistillata* to elucidate either i) the importance of lifestyle and adaptation potential, and ii) the role of the genomic equipment and niche adaptation potential. Six complete genomes were *de novo* hybrid assembled using a combination of short and long-read technologies. These *Paracoccus* genomes increase the number of completely closed high-quality genomes of type strains from 15 to 21. Pan-genome analysis revealed an open pan-genome composed of 13,819 genes with a minimal chromosomal core (8.84%) highlighting the genomic adaptation potential and the huge impact of extra-chromosomal elements. All genomes are shaped by the acquisition of various mobile genetic elements including genomic islands, prophages, transposases, and insertion sequences emphasizing their genomic plasticity. In terms of lifestyle, each mobile genetic elements should be evaluated separately with respect to the ecological context. Free-living genomes, in contrast to host-associated, tend to comprise (1) larger genomes, or the highest number of extra-chromosomal elements, (2) higher number of genomic islands and insertion sequence elements, and (3) a lower number of intact prophage regions. Regarding lifestyle adaptations, free-living genomes share genes linked to genetic exchange via T4SS, especially relevant for *Paracoccus*, known for their numerous extrachromosomal elements, enabling adaptation to dynamic environments. Conversely, host-associated genomes feature diverse genes involved in molecule transport, cell wall modification, attachment, stress protection, DNA repair, carbon, and nitrogen metabolism. Due to the vast number of adaptive genes, *Paracoccus* can quickly adapt to changing environmental conditions.

(CP067169-CP067178) and raw reads have been deposited in the NCBI SRA database (SRR23190560 and SRR23190561). P. alcaliphilus DSM 8512T under BioProject PRJNA689372, BioSample SAMN17207692, accession numbers (CP067124-CP067128) and raw reads have been deposited in the NCBI SRA database (SRR23190624 and SRR23190625). P. fistulariae KCTC 22803T under BioProject PRJNA689404, BioSample SAMN17209169, accession numbers (CP067136- CP067139) and raw reads have been deposited in the NCBI SRA database (SRR23190620 and SRR23190621). P. saliphilus DSM 18447T under BioProject PRJNA689405, BioSample SAMN17209211, accession numbers (CP067140-CP067141) and raw reads have been deposited in the NCBI SRA database (SRR23190618 and SRR23190619). P. seriniphilus DSM 14827T under BioProject PRJNA689377, BioSample SAMN17208906, accession numbers (CP067129-CP067133) and raw reads have been deposited in the NCBI SRA database (SRR23190622 and SRR23190623). P. stylophorae LMG 25392T under BioProject PRJNA689381, BioSample SAMN17208933, accession numbers (CP067134-CP067135) and raw reads have been deposited in the NCBI SRA database (SRR23190562 and SRR23190563). All numbers are depicted in S1 Table.

**Funding:** This study was partly supported by the Deutsche Forschungsgemeinschaft (DFG) as part of the collaborative research center TRR51 Roseobacter awarded to TB and RD. We acknowledge support by the Open Access Publication Funds of the Göttingen University. The funders had no role in study design, data collection and analysis, decision to publish, or preparation of the manuscript.

**Competing interests:** The authors have declared that no competing interests exist.

# Introduction

The genus *Paracoccus* is the type genus of the *Paracoccaceae* [1], formerly member of the *Rhodobacteraceae*, which is known for a broad spectrum of ecologically relevant metabolic traits and high abundance in many marine environments [2]. *Paracoccus* seems to be a biogeographic cosmopolite due to its distribution all over the world and inhabiting a variety of different ecological niches [2]. Nevertheless, despite its wide distribution, their role in the marine biogeochemical cycles and their ability to inhabit a wide variety of different environments remains to be elucidated. In addition, different lifestyles are described for *Paracoccus*, such as free-living, host-associated, pathogenic, or symbiotic [3–5]. Furthermore, high potential for genomic adaptability is a prerequisite for their global omnipresence. Therefore, genomic analysis of *Paracoccus* should focus on horizontal gene transfer (HGT) and mobile genetic elements (MGEs) including plasmids, genomic islands (GIs), insertion sequences (IS) elements, transposases and prophages since these are drivers of adaptation and evolution in prokaryotes [6]. So far 84 *Paracoccus* species were validly published (https://www.ezbiocloud.net/search?tn=paracoccus accessed 2022-09-22). However, the majority of newly announced *Paracoccus* species are taxonomically classified by 16S rRNA gene analysis, which is known to be insufficient for a robust species classification and often lead to unreliable or even wrong taxonomic assignment [7]. Correspondingly, several genera and species within the family *Rhodobacteraceae*, were reclassified [8]. Furthermore, a discrepancy in the taxonomic framework was detected by core-pan genome analysis of the genus *Paracoccus* [9]. Notably, 64 *Paracoccus* type strains genomes are available, but only 15 have a complete status. Complete genome sequences are critical for all downstream analysis including comparative genomics, phylogenetic analysis, and functional annotation or inference of biological relationships. Thus, more complete high-quality *Paracoccus* genomes are required to provide a solid basis for robust taxonomic assignment of new *Paracoccus* isolates. To widen our understanding of how *Paracoccus* is able to adapt to various environments and develop different lifestyles, we provide six high-quality *Paracoccus* genomes of the type strains *P. aestuarii* DSM 19484$^T$ (= B7$^T$, = KCTC 22049$^T$) [10], *P. alcaliphilus* DSM 8512$^T$ (TK 1015$^T$ = JCM 7364$^T$) [11], *P. fistulariae* KCTC 22803$^T$ (= 22-5$^T$, = CGUG 58401$^T$) [12], *P. saliphilus* DSM 18447$^T$ (= YIM 90738$^T$, = CCTCC AB 206074$^T$) [13], *P. seriniphilus* DSM 14827$^T$ (= MBT-A4$^T$, = CIP 107400$^T$) [14], and *P. stylophorae* LMG 25392$^T$ (= KTW-16$^T$, = BCRC 80106$^T$) [15]. Strains were chosen to represent different habitats including tidal flat sediment [10], soil [11], bluespotted cornetfish *Fistularia commersonii* [12], saline-alkaline soil [13], marine bryozonan *Bugula plumosa* [14], and the reef-building coral *Stylophora pistillata* [15], respectively. The six strains can be assigned to the two lifestyles free-living and host associated. All genomes were sequenced using short-read (Illumina) and long-read (Oxford Nanopore) technology. Subsequently, genomes were *de novo* hybrid assembled and phylogenetically grouped into the genus *Paracoccus* by using all genomes available in the RefSeq database [16]. Thereby, the quality of all genomes was evaluated, followed by downstream analyses that involved determination of the pan-genome, functional annotation, and genome comparison. The latter included identification and comparison of plasmids, IS elements, GIs, transposases, and (pro)phages. This detailed comparative investigation of six *Paracoccus* genomes and their functional repertoire will help shading light on how *Paracoccus* is able to distribute globally and adapt to ecological niches.

# Results and discussion

## Genome features of *Paracoccus* type strains

As of September 2022, 185 *Paracoccus* genomes (64 type/representative strains) were available in the genome databases of which only 15 had the quality status "complete". We increased the

**Table 1. Genomic features of the *Paracoccus* strains presented in this study.**

| Strain | Genome Size (bp) | Extrachromosomal Elements | GC % | CDS | gene | rRNA | tmRNA | tRNA |
|---|---|---|---|---|---|---|---|---|
| *P. aesturarii* DSM 19484[T] | 3,675,213 | 9 | 67.8 | 3609 | 3701 | 9 | 1 | 56 |
| CP067169 | 3,089,821 | | 67.8 | 3048 | 3136 | 9 | 1 | 55 |
| CP067170 | 296,221 | | 68.72 | 281 | 281 | - | - | - |
| CP067171 | 190,534 | | 69.57 | 181 | 185 | - | - | 1 |
| CP067172 | 60,130 | | 63.74 | 68 | 68 | - | - | - |
| CP067173 | 10,648 | | 58.87 | 9 | 9 | - | - | - |
| CP067174 | 9,146 | | 57.65 | 6 | 6 | - | - | - |
| CP067175 | 5,850 | | 58.62 | 8 | 8 | - | - | - |
| CP067176 | 5,435 | | 51.72 | 2 | 2 | - | - | - |
| CP067177 | 4,502 | | 57.93 | 4 | 4 | - | - | - |
| CP067178 | 2,926 | | 64.42 | 2 | 2 | - | - | - |
| *P. alcaliphilus* DSM 8512[T] | 4,772,712 | 4 | 64.3 | 4667 | 4757 | 6 | 1 | 52 |
| CP067124 | 3,567,425 | | 64.2 | 3573 | 3654 | 6 | 1 | 48 |
| CP067125 | 429,813 | | 64.98 | 393 | 396 | - | - | 1 |
| CP067126 | 402,633 | | 64.76 | 378 | 382 | - | - | 2 |
| CP067127 | 305,818 | | 64.1 | 244 | 245 | - | - | - |
| CP067128 | 77,023 | | 60.42 | 79 | 80 | - | - | 1 |
| *P. fistulariae* KCTC 22803[T] | 3,723,810 | 3 | 63.1 | 3599 | 3678 | 6 | 1 | 51 |
| CP067136 | 3,621,100 | | 63.12 | 3497 | 3575 | 6 | 1 | 51 |
| CP067137 | 90,204 | | 61.01 | 86 | 87 | - | - | - |
| CP067138 | 7,465 | | 59.89 | 11 | 11 | - | - | - |
| CP067139 | 5,041 | | 58.88 | 5 | 5 | - | - | - |
| *P. saliphilus* DSM 18447[T] | 4,621,331 | 1 | 65.1 | 4411 | 4487 | 6 | 1 | 47 |
| CP067140 | 4.,378,448 | | 61.11 | 4179 | 4251 | 6 | 1 | 47 |
| CP067141 | 242,883 | | 60.79 | 232 | 236 | - | - | - |
| *P. seriniphilus* DSM 14827[T] | 4,261,722 | 4 | 61.5 | 4007 | 4092 | 9 | 1 | 50 |
| CP067129 | 2,779,753 | | 61.83 | 2666 | 2740 | 6 | 1 | 45 |
| CP067130 | 599,725 | | 61.36 | 555 | 556 | - | - | - |
| CP067131 | 449,297 | | 60.52 | 398 | 399 | - | - | - |
| CP067132 | 420,573 | | 61.06 | 374 | 383 | - | - | - |
| CP067133 | 12,374 | | 57 | 14 | 14 | - | - | - |
| *P. stylophorae* LMG 25392[T] | 3,642,172 | 1 | 66.7 | 3521 | 3598 | 6 | 1 | 47 |
| CP067134 | 3,634,760 | | 66.74 | 3515 | 3592 | 6 | 1 | 47 |
| CP067135 | 7,412 | | 55.38 | 6 | 6 | - | - | - |

number of complete high-quality type strain genomes to 21 by sequencing and analyzing the genomes of six *Paracoccus* type strains. The strains originate from various habitats including soils, sediments, and associated to diverse hosts. All following genome statistics including short- and long-read sequencing statistics are summarized in S1 Table. Complete genomes were hybrid assembled from Oxford Nanopore and Illumina reads. The sequencing led to a 148–1,320x genome coverage and resulted in one circular chromosome per type strain and multiple extrachromosomal circular elements ranging from 1 to 9. (Table 1; S1 Table).

The total genome sizes ranged from 3.64 to 4.77 Mbp, with an average GC content of 61.5 to 67.8%. Each strain encoded one tm-tRNA, harbors 6–9 rRNA and 47–56 t-RNA molecules. The quality of the genomes was assessed and evaluated with BUSCO v.5.4.5 [17] (Fig 1) and initial phylogenetic classification was performed using the Genome Taxonomy Database Toolkit (GTDB-Tk v2.1.1) [18] summarized in S2 Table.

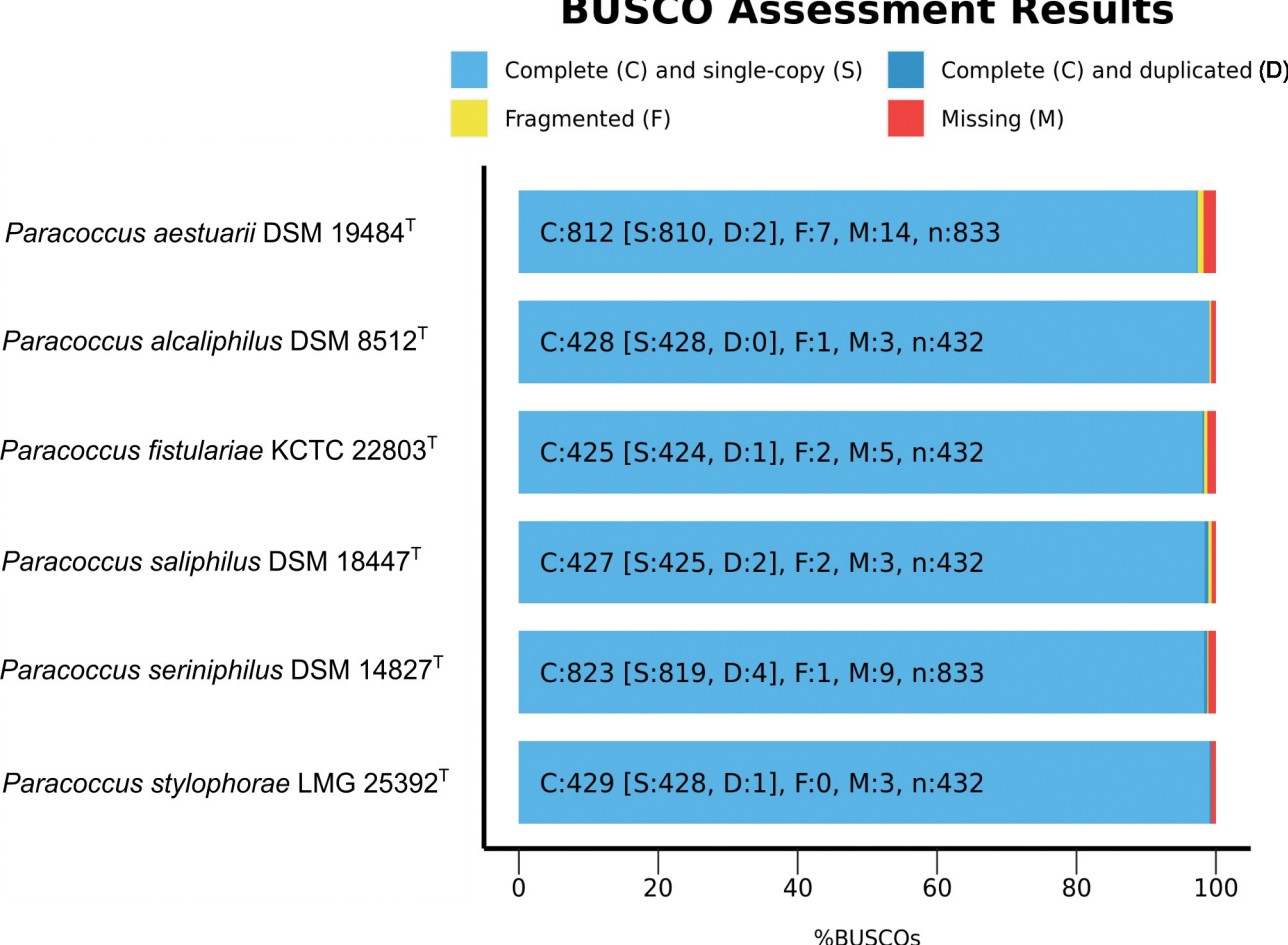

**Fig 1. Genome assembly quality assessment and evaluation of the six *Paracoccus* type strains.** The chart was generated with BUSCO v.5.4.5 showing the relative completeness of each strain's genome.

BUSCO has identified all isolates to the family *Rhodobacteraceae*, now reclassified to a separate family *Paracoccaceae* [1]. The initial phylogenetic classification assigned all genomes to their correct species group (S2 Table). However, none of the bioinformatic tools accounted the new family level *Paracoccaceae* [8]. Nevertheless, all genomes included almost all single copy genes (> 97.48%).

## Phylogeny of *Paracoccus*

Quality control of all available *Paracoccus* genomes including our six novel genomes (n = 182) revealed that all type strains have a completeness >97% with a contamination rate <2% excluding *P. mutanolyticus* RSP-02 which was therefore excluded (S3 Table). Moreover, 21 *Paracoccus* strains were excluded because they did not meet the aforementioned completeness or contamination threshold. In total, 160 genomes were implemented in the phylogenetic analysis of the genus *Paracoccus* including our six type strain genomes (Fig 2 and S4 Table).

We detected 16 novel classifications and five misclassifications within the genus *Paracoccus* (Table 2 and S4 Table). Some of the misclassification were already detected by Puri et al, 2021 [9]. However, this analysis highlights the importance of high-quality reference genomes to establish a robust taxonomic classification as novel *Paracocci* were misclassified at the end of

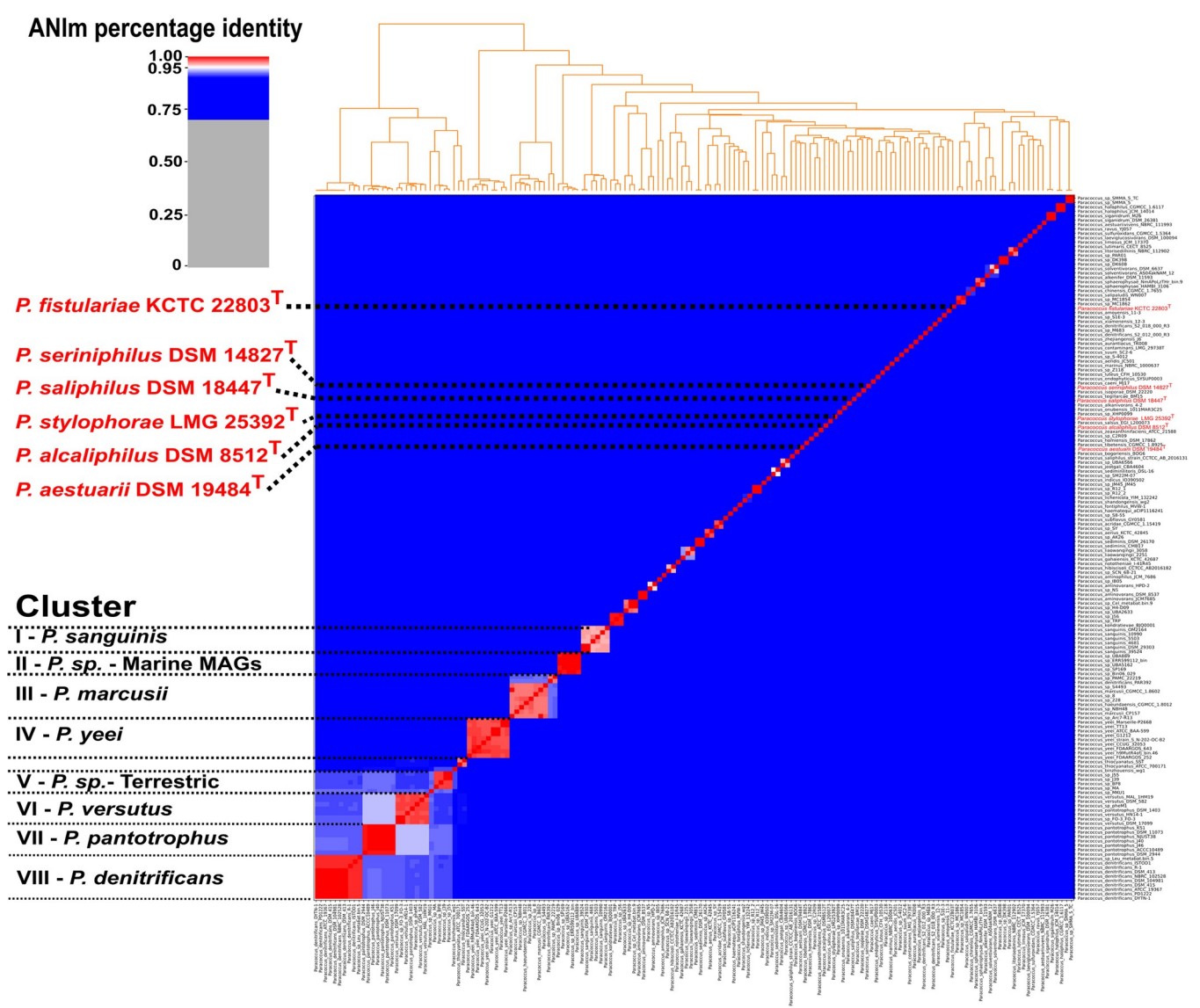

**Fig 2. Phylogenetic analysis of the genus *Paracoccus*.** Type strains and the 8 largest *Paracoccus* clusters are highlighted in red and black. Detailed output of all available *Paracoccus* genomes including ANI values is summarized in S4 Table. Modifications were performed with Inkscape v 1.2. [19] (1. Inkscape Project. Inkscape [Internet]. 2020. Available from: https://inkscape.org).

2022 (Table 2, underlined) or for new subjected *Paracocci* genomes a classification was simply not performed.

*Paracoccus denitrificans* (formerly known as *Micrococcus denitrificans*) was the first described *Paracoccus* species and represent the largest cluster (cluster VIII; Fig 2) [20]. Cluster III (*P. marcusii)* and cluster IV (*P. yeeii*) are represented by nine strains. Notably, most of the unclassified *Paracoccus* sp. strains were associated to cluster III (Fig 2, Table 2, S4 Table). Moreover, as suggested by Leinberger et al. (2021) [21] the species *P. haeundaenis* should be resolved into the species *P. marcusii* which was first described in 1998 [22]. *P. haeundaensis* was reported in 2004 [23] as novel species based on 16S rRNA gene profiling. Often 16S rRNA gene sequencing is inconclusive for species level classification, especially for members of the family *Rhodobacteraceae* [8] which recently lead to a new family announcement of

**Table 2. Reclassification of *Paracoccus sp*. and wrongly assigned *Paracoccus*.**

| New species classifications | Database Strain Assignment | Corrected Classification | Closest type/representative strain | ANIm |
|---|---|---|---|---|
| 1 | *P. sp* UBA6566 | *P. jeotgali* UBA6566 | *P. jeotgali* CBA4604 | 96.2% |
| 2 | *P. sp* SM22M-07 | *P. sediminilitoris* SM22M-07 | *P. sediminilitoris* DSL-16 | 95.1% |
| 3 | *P. sp* SY | *P. acridae* SY | P. acridae CGMCC 1.15419 | 98.1% |
| 4 | *P. sp* AK26 | *P. aerius* AK26 | P. aerius KCTC 42845 | 98.4% |
| 5 | *P. sp* SCN 68–21 | *P. hibiscisoli* SCN 68–21 | *P. hibiscisoli* CCTCC AB2016182 | 96.3% |
| 6 | *P. sp* J56 | *P. kondratievae* J56 | *P. kondratievae* BJQ0001 | 99.4% |
| 7 | *P. sp* TRP | *P. kondratievae* TRP | *P. kondratievae* BJQ0001 | 99.5% |
| 8 | *P. sp* NBH48 | *P. marcusii* NBH48 | *P. marcusii* CP157 | 97.8% |
| 9 | *P. sp* 228 | *P. marcusii* 228 | *P. marcusii* CP157 | 97.7% |
| 10 | *P. sp* S4493 | *P. marcusii* S4493 | *P. marcusii* CP157 | 97.5% |
| 11 | *P. sp* 8 | *P. marcusii* 8 | *P. marcusii* CP157 | 97.4% |
| 12 | *P. sp* Arc7-R13 | *P. marcusii* Arc7-R13 | *P. marcusii* CP157 | 97.5% |
| 13 | *P. sp* FO-3 | *P. versutus* FO-3 | *P. versutus* DSM 582 | 98.6% |
| 14 | *P. sp* pheM1 | *P. versutus* pheM1 | *P. versutus* DSM 582 | 98.6% |
| 15 | *P. sp* Leu metabat.bin.5 | *P. denitrificans* Leu metabat.bin.5 | *P. denitrificans* DSM 413 | 98.9% |
| 16 | *P. sp* PAR01 | *P. litorisediminis* PAR01 | *P. litorisediminis* NBRC 112902 | 97.5% |
| **Reclassification** | | | | |
| 17 | *P. haeundaensis* CGMCC 1.8012 | *P. marcusii* CGMCC 1.8012 | *P. marcusii* CP157 | 97.6% |
| 18 | *P. aminovoroans* HPD-2 | *P. sp* HPD-2 | *P. aminovorans* DSM 8537 | 89.8% |
| 19 | *P. denitrificans* PAR392 | *P. sp* PAR392 | *P. marcusii* CGMCC 1.8602 | 92.1% |
| 20 | *P. pantotrophus* DSM 1403 | *P. versutus* DSM 1403 | *P. versutus* DSM 582 | 98.6% |
| 21 | *P. denitrificans* S2 018 000 R3 | *P.sp.* S2 018 000 R3 | *P. yeeii* ATCC BAA-599 | 84.8% |

*Paracoccacea* [1]. Cluster IV (*P. yeeii*) and Cluster I (*P. sanguinis*) comprise strains described as opportunistic human pathogens, which is a unique feature within the genus *Paracoccus* [24, 25]. Two closely related clusters are cluster VI (*P. versutus*) and cluster VII (*P. pantotrophus*). Both type strains showed an identity value of 93.4%, which is close to the species boundary of ~94–96% which was proposed by Richter et al 2009 [26] and Konstantinidis et al. 2005 [27]. Considering that even the average nucleotide identity value between these two *Paracoccus* species is so close, it becomes clear that the usage of the 16S rRNA gene would be insufficient and error-prone for new strain classifications. Already misclassified strains such as *P. pantotrophus* DSM 1403[T] had an ANI value of 98.6% to the type strain *P. versutus* DSM 582[T] while, only 93.6% to the type strain *P. pantrotophus* DSM 2944[T] [9]. In the future, valid classification of new genomes will become even more important as data generated by metagenome sequencing allows the assembly of metagenome assembled genomes (MAGs), which could lead to rapid error propagation within databases due to omitted or insufficient performed phylogeny.

## Pan-genome analysis of *Paracoccus*

The pan- and core-genome of the six *Paracoccus* type strains followed the general results of the genus *Paracoccus* [9] for which an open pan-genome is proposed. The fitting least-squares curve based on Heaps' Law indicates an open pan-genome ($\alpha = 0{,}3750183$) (Fig 3A and 3B).

The core size accumulation curve saturates after two genomes, which suggests a stable minimum core. In our analysis of protein-encoding genes, we found that the *Paracoccus* pan-genome consisted of a total of 13,819 gene clusters. Interestingly, most of these gene clusters were present in the cloud genome (10,634 gene clusters, 76,95%) compared to the core (1,221

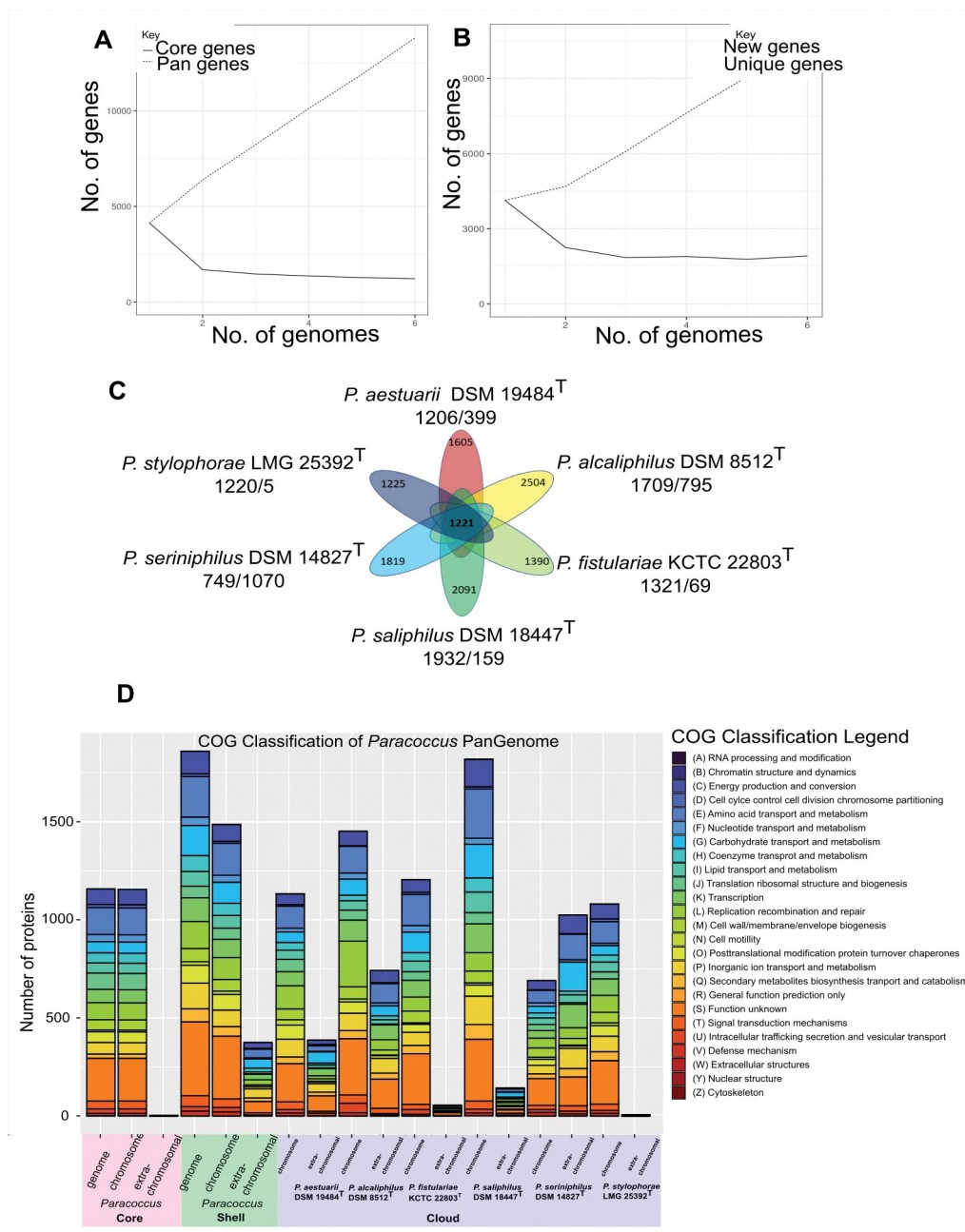

**Fig 3. Pan-core analysis of six *Paracoccus* type strain genomes.** A) Pan- and core size accumulation, and B) Cloud and new genes accumulation in the analyzed *Paracoccus* genomes. C) Distribution of core (center) and cloud (ellipses) gene clusters in pan-genome. The distribution of cloud gene clusters located on the chromosome or extrachromosomal elements is listed under each type strain (chromosome/extrachromosomal element). D) Bar charts of the pan-genome functional classification annotated in COG databases. The function was plotted either according to core (light red, left), shell (light green) and cloud (light blue, green) and according to their genomic localization, chromosomal or extrachromosomal.

gene clusters, 8,84%) and shell (1,964 gene clusters, 14.21%) genome (Fig 3D, S5 Table). These results emphasize the highly adaptive genome and non-conserved number of genes across the *Paracoccus* genus.

To assess the level of functional diversity within the *Paracoccus* type strain core, shell and cloud genomes, COG analysis was employed (Fig 3D). The COG category analysis revealed clusters in the *Paracoccus* core genome were involved in a high amount of class (S) function unknown, (E) amino acid transport and metabolism, (L) replication, recombination, and repair, (J) translation ribosomal structure and biogenesis, and (C) energy production and conversion nearly exclusively encoded on the chromosome. Genes composing the shell genome were mainly from class E (amino acid transport and metabolism), followed by class (G) carbohydrate transport and metabolism, (G) carbohydrate transport and metabolism, (L) replication, recombination, and repair, (P) inorganic ion transport and metabolism, (K) transcription, and (C) energy production and conversion, where ~20% are encoded by extra-chromosomal elements. The functional composition of the cloud is not particularly different from that of the core with the main groups (E, L, G, K P). This is in line with previous studies, which showed same functional key groups for core and cloud of *Paracoccus yeei* CCUG 32053 [5]. However, the analysis of chromosomal and extra-chromosomal core or cloud genes revealed that the functional classes are generally similar, but the key groups are specific for the chromosome and extra-chromosomal elements. Additionally, as illustrated in Fig 3C and 3D, a substantial portion of the genes (up to58.82% for *P. seriniphilus* DSM 14827[T]) are located on extra-chromosomal elements. These genes are involved in processes such as amino acid transport and metabolism, replication, recombination and repair, carbohydrate transport and metabolism, transcription, and inorganic ion transport and metabolism (Fig 3D). These results highlighting the role of specific functions gained by transmission through horizontal gene transfer that enable adaptation and survival in a specific ecological niche also supported by a recent study that showed a significant enrichment of functional groups specific for chromosomes versus extra-chromosomal elements in the human gut [28]. However, to further investigate genetic differences between the free-living and host-associated *Paracoccus* genomes we inspected shared genes specific for each lifestyle (S6 Table). In total 22 specific in free-living (FL) and 38 genes specific for host-associated (HA) genomes were identified (S6 Table). Notably, 50% of FL genes are *trb*-genes and genes involved in type IV secretion system (T4SS). Genes of this cluster are known for enabling transfer of genetic material especially plasmids [29]. In context of a free-living lifestyle where the environment and nutrient availabilities change fast a maintenance of a system for genetic material exchange for acquisition of new genetic traits is plausible. *P. aestuarii* DSM19484[T] and *P. alcaliphilus* DSM8512[T] encode those genes chromosomally, in contrast in *P. saliphilus* DSM18447[T] these genes are found on the only plasmid (S6 Table, extra-chromosomal location highlighted in green). Besides, two genes (*nasE*, *nasD*) involved in nitrite assimilation and conversion were identified indicating a role in the nitrogen cycle and energy metabolism. Nitrite reductases are enzymes catalyzing the reduction of nitrite to ammonia [30]. In combination, a gene involved in the transport of glutamine (*gln*Q) was identified which is in general described with glutamate and ammonium as preferred source for nitrogen since the conversion of nitrate to ammonium is a high-energetic process [31]. Our data reveal an increased genomic equipment and resulting traid for biochemical cycling of nitrogen of free-living *Paracocci*.

In contrast, in host-associated genomes the majority of genes are associated to various transporters (*oppF*, *dppD*, *opuE*, *hmuU*, *dctM*, *dctQ*, *tsaT*, *vexB*), regulators (*gmuR*, *hpaR*), in attachment and motility (*fliC*, *flaF*, *ufaA*), as well as carbon/nitrogen cycling (formamidase), indicating adaptations to the host by cell adhesion, substrate exchange, and host interaction. Broadly, an elevated abundance of genes observed in HA genomes predominantly comprises transporter genes including members of the ABC family [32], which may be pivotal in mediating substrate exchange and host interactions [5]. Notably, transporters have been also predicted in the opportunistic pathogen *P. yeei*, as virulence factor [5]. This is particularly

significant in contexts involving HA nutrient scarcity, competition, and host-specific nutrient acquisition. Furthermore, transporters could serve dual roles, including functioning as efflux pumps to fortify the bacterium's defense mechanisms within the host environment [33]. Other adaptation to hosts could be seen in cell wall modification or optimization of attachment. In HA genomes, biotine carboxylase was found which participates in fatty acid biosynthesis which could be essential for cell membrane modification [34] as well as tuberculostearic acid methyltransferase (*ufaA*) which modifies fatty acids and may help to adhere to the host. HA bacteria are often faced with stresses coursed by the host. An adaptation of HA *Paracocci* may be the lysine decarboxylase to regulate intracellular pH, by building cadaverine, which are known from *Paracoccus* and other marine bacteria to tolerate changes in pH and salinity [35–37]. Additionally, thiamine-phosphate synthase and thymidylate synthase 2 were found. The thiamine-phosphate synthase is involved in vitamin B1 biosynthesis, by the conversion of 4-Amino-5-hydroxymethyl-2-methylpryriminide diphosphate into thiamine phosphate. Vitamin B1 is an essential cofactor for bacteria [38] so HA environments may lead to limitations which are circumvented by *de novo* synthesis. In the same way, thymidylate synthase 2 (*thyX*) ensures the synthesis, replication and repair of DNA by the conversion of dUMP to dTMP [39]. Lastly, a formamidase was identified in all HA genomes which hydrolysis formamide into formate an ammonia. *Paracoccus* is known as a formate-utilizing bacterium an both products are either used as carbon or nitrogen source [40].

## Comparative genomics of the genus *Paracoccus*

Genome reduction was once thought to be a distinctive feature of endosymbiotic bacteria that live inside host cells, either as mutualists or obligate pathogens [41]. However, it has been observed in various free-living bacteria as well [42]. We observed a similar trend for our investigated strains where in general free-living strains tend to have larger genomes. However, *P. aestuarii* DSM 19484[T] comprised the smallest genome but the highest number of extrachromosomal elements (Table 1). To gain insights into the adaptability and evolutionary potential of *Paracoccus*, the six type strain genomes were screened for mobile genetic elements (MGEs) such as genomic islands (GI), insertion sequences (IS), transposases, and prophages (P), as well as genes involved in the synthesis of secondary metabolites (SMs). All genomes contained a diverse array of MGEs and SM gene clusters (Tables 3 and 4; detailed output in S7 Table), highlighting their genomic plasticity and potential for adaptation and evolution in various environments.

Each type strain presented here represents a snapshot of adaptation to a specific ecological niche. Three strains can be grouped based on their lifestyles as either free-living (Table 3, grey)

**Table 3. Number of detected MGEs and SMs in the analyzed *Paracoccus* genomes.**

| Strain | Number of islands | Number of IS elements | Number of prophage regions | Number of potential secondary metabolites cluster |
|---|---|---|---|---|
| *P. aestuarii* DSM 19484[T] ([1]) | 14 | 113 (3.1 %) | 3 (incomplete) | 11 |
| *P. alcaliphilus* DSM 8512[T] ([2]) | 23 | 247 (6.29 %) | 15 (13 incomplete;2 questionable) | 8 |
| *P. fistulariae* KCTC 22803[T] ([3]) | 13 | 48 (1.40 %) | 3 (2 incomplete; 1 questionable) | 8 |
| *P. saliphilus* DSM 18447[T] ([4]) | 21 | 81 (1.90 %) | 3 (incomplete) | 8 |
| *P. seriniphilus* DSM 14827[T] ([5]) | 13 | 78 (2.47 %) | 8 (1intact, 4 incomplete, 3 questionable) | 7 |
| *P. stylophorae* LMG 25392[T] ([6]) | 9 | 45 (2.28 %) | 10 (3 intact; 4 incomplete, 3 questionable) | 8 |

or host-associated. The free-living strains possess a higher number of GIs (14–23) and IS elements (81–247) compared to host-associated strains (GI: 9–13; IS: 45–78). This is consistent with a study by Newton and Bordenstein [43], who found that extracellular (free-living) bacteria tend to have a higher number of MGEs compared to obligate intracellular (host-associated) bacteria, likely due to the more variable and challenging environments they encounter. This trend was also observed in *Paracoccus*, although it should be considered that the here presented host-associated strains are able to survive free-living and therefore do not completely correspond to the definition of an obligate intracellular organism.

Genomic islands and prophages have been identified in several species of *Paracoccus* suggesting genome adaption by conferring fitness or virulence to their hosts [5, 44]. While it has been established that prophages can facilitate the acquisition of antibiotic resistance by their hosts [45], the frequency of such events is lower than initially anticipated during the recent surge in phage research [46]. To date, no phages have been identified in *Paracoccus* that carry genes associated with either resistance or virulence but potential genes contributing to fitness are documented [44]. We conducted an analysis of all the genes present in the genomic islands and putative prophage regions of the six type strains, focusing on their potential association in conferring either fitness or virulence (Table 4).

The majority of genes identified in the genomic islands of the six type strains were categorized as hypothetical or were characteristic with other mobile genetic elements such as IS sequences and transposases. Nevertheless, each strain harbored genes associated with increased fitness, including those encoding type secretion systems (TSS), toxin-antitoxin systems, antibiotic resistance, and heavy metal transporters. Additionally, genes associated with hemolysin and phospholipase were also identified, indicating the potential for virulence in these bacteria [47, 48]. The presence of T1SS and T4SS genes in *Paracoccus* implies their significance in enhancing substrate and toxin transport and facilitating exchange of DNA and proteins between bacteria. In addition, the identification of multiple Type II and Type IV toxin-antitoxin (TA) system genes suggests their significance in conferring survival under stressful conditions, such as nutrient deprivation, exposure to antibiotics, or phage infection [44]. Moreover, bacterial populations can be stabilized through the elimination of cells carrying low amounts of MGE [49]. The identification of β-lactam and streptogramin antibiotic resistance genes, as well as various heavy metal transporters, such as cadmium, mercury, and arsenate, highlights their importance for the uptake and detoxification of heavy metals in specific contaminated ecological niches. Especially in the genus of *Paracoccus*, phages seem to be species specific and confer metal resistances tailored to the habitat [44]. The identification of methyl-transferases (MTases) and restriction modification (RM I or RM III) associated genes in all *Paracoccus* strains emphasizes the diverse array of mechanisms available to *Paracoccus* to protect against foreign DNA, plasmids, and phage attacks or to enhance fitness by acquiring new restriction enzymes for utilizing DNA from a new source [50]. Additionally, methyltransferases facilitate the modification of own DNA, thereby enabling better survival in a specific environment. Our analysis showed no clear effect of the lifestyle on MGE equipment. However, a higher number of intact prophages was observed in the host-associated strains *P. seriniphilus* DSM 14827[T] and *P. stylophorae* LMG 25392[T] (S7C Table). One possible explanation is that *Paracocci* that are associated with a host are generally subjected to unique selective pressures specific to the host, which can lead to a higher frequency and maintenance of prophages spreading virulence and fitness factors to their bacterial host [51]. In contrast, prophages can be easily acquired and lost by free-living bacteria that are not subjected to host-specific pressures [51]. However, these bacteria must also contend with the varying and often unfavorable environmental conditions of the open ocean [51]. Furthermore, the bacterial adaptation mechanism prioritizes the fixation of phage genes that confer benefits to the host, while genes that

**Table 4. Detailed summary of detected fitness and virulence associated genes located in GI — genomic islands, P–prophage, and extrachromosomal.**

| Strain | 1 | 2 | 3 | 4 | 5 | 6 |
|---|---|---|---|---|---|---|
| **Fitness factors** | | | | | | |
| **Types of secretion system associated genes** | | | | | | |
| I | - | - | - | 2 | - | - |
| IV | 1 | 1 | 1 | - | - | - |
| **Toxin antitoxin system associated genes** | | | | | | |
| II | 2; 2 | 4; 1 | 3, 3 | 2 | - | 7<br>3 |
| IV | 2 | 1; 1<br>1 | - | - | - | - |
| **Antibiotic resistance associated genes** | | | | | | |
| b-lactame | - | - | - | 1 | 1<br>1 | - |
| aminoglycoside | - | - | - | - | - | - |
| tetracycline | - | - | - | - | - | - |
| acyltranferase chlorampenicol | - | - | - | - | - | - |
| streptogramin | - | - | - | - | 1 | - |
| **Heavy metal resistance associated genes** | | | | | | |
| cadmium | 1; 1 | | 1 | - | - | 5 |
| copper | 2; 1 | 2 | 4; 1 | 1; 1 | - | 6<br>1 |
| mercury | - | - | 4 | - | - | 2<br>2 |
| zinc | - | - | 1 | - | - | 3 |
| arsenate | 1 | 2 | - | - | - | 3 |
| chaperone | 2; 1 | - | - | 1 | | 2 |
| proteases | 1 | - | - | 1 | 1 | 3<br>1 |
| nitrogen fixation | - | - | - | - | - | 1 |
| nitrate reductase | 2 | - | - | - | - | |
| cytochrome | - | 1; 3 | 2; 1 | - | - | 3 |
| polyhydroxyalkanoatesynthase | 1 | 1 | 1 | - | 1 | |
| nitrite hydratase | - | - | - | - | - | 1 |
| **Virulence factors** | | | | | | |
| hemolysin | - | - | 1 | - | - | - |
| phospholipase | 1 | - | - | - | - | - |
| **Phage associated structural genes** | | | | | | |
| structural genes | 7 | 7 | 14 | 6 | 8 | 26 |
| restriction modification systems | 2 | 1 | - | 2 | 2 | 1 |
| I | - | - | 2 | 1 | 2; 3 | 6<br>2 |
| III | 1 | - | - | - | - | 1 |
| methyltransferasen | 1; 1 | 5; 1<br>5 | 5<br>1 | 1; 5<br>1 | 1 | 8<br>8 |

lack selective value gradually decay over time, accounting for the incomplete and uncertain detection of phages [51]. More IS elements were detected in type strains which are free-living (Table 3, Fig 4).

Our analysis revealed 612 insertion sequence (IS) elements representing 20 IS families (Fig 4). Most abundant families were IS5 and IS3 while other IS families are unevenly

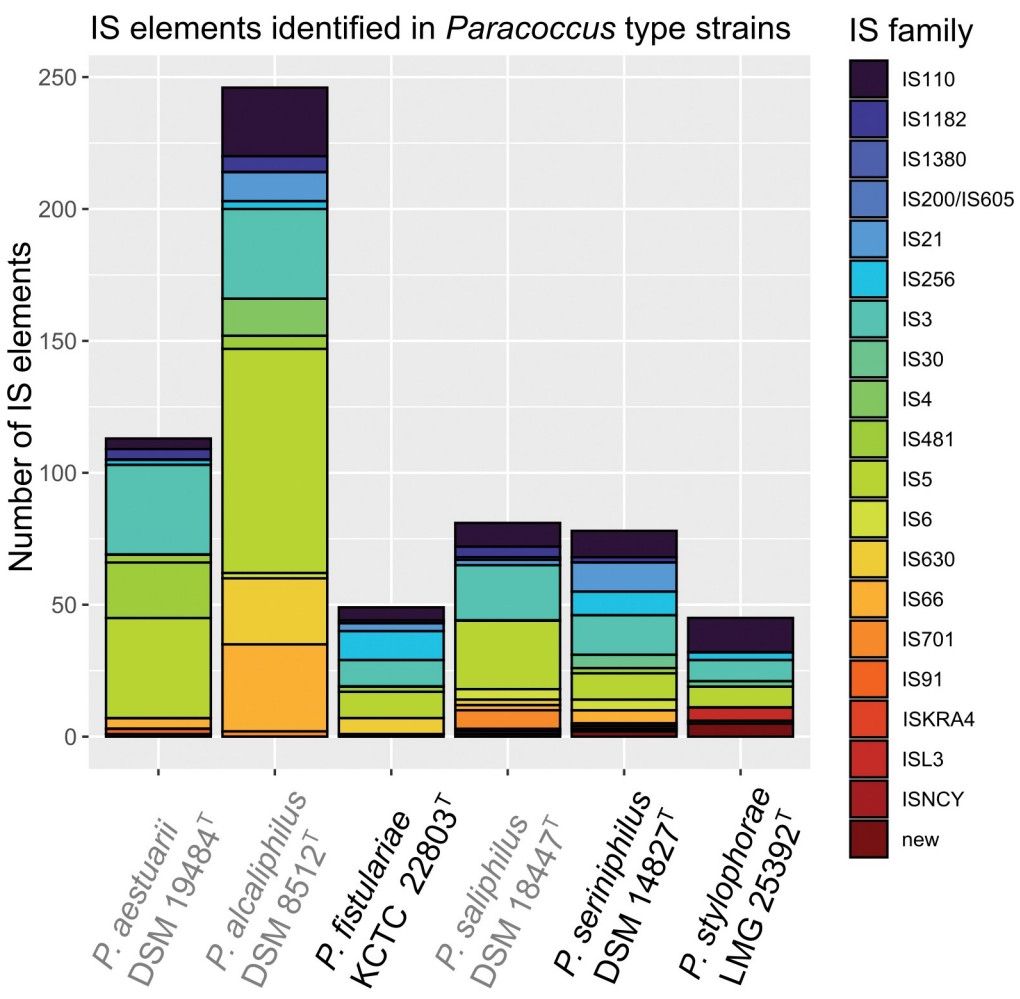

**Fig 4. Distribution of IS-families identified in the analyzed *Paracoccus* type strains.** The elements are members of 20 IS families represented by a different color. Free-living associated strains strains are displayed in grey. Family IS481, IS4, IS1380, IS200/IS605, and ISKRA4 are unique to *P. aestuarii* DSM19484[T], *P. alcaliphilus* DSM 8512[T], *P. fistulariae* KCTC 22803[T], *P. saliphilus* DSM 18447[T], and *P. seriniphilus* DSM14827[T], respectively.

distributed in *Alphaproteobacteria* [52] and in the ISfinder database [53]. Notably, Dziewit et al. 2012 described only 10 IS families in the mobilome of a *Paracoccus* spp. lacking families IS1, IS3, IS91 and IS110 which are common in other *Alphaproteobacteria* [52], which might be due to the rapid increase of new identified IS families [54]. In *P. yeei* only seven IS families (IS3, IS5, IS30, IS66, IS256, IS110 and IS1182) were identified [5]. Our strains include eight up to 14 different IS elements where IS110, IS21, IS66 and IS256 made the majority, whereas IS*As1* described as rare in *Paracoccus* were not identified [52]. Additionally, we identified rare strain specific IS elements such as IS481 (*P. aestuarii* DSM 19484T), IS4 (*P. alcaliphilus* DSM 8512[T]); IS1380 (*P. fistulariae* KCTC 22803[T]); IS200/IS605 (*P. saliphilus* DSM 18477[T]), and ISKRA4 (*P. seriniphilus* DSM 14827[T]) (Fig 4). Host associated strains had the lowest amount of IS which is in line with the theory that a restricted environment for example on a host act as a bottleneck, where the specific nutrient richness of a host can lead to a genome streamlining. However, in free-living bacteria IS elements thought to be more beneficial in fluctuating

environments which allow a higher degree of genetic plasticity for rapid acquisition of new traits [53]. Since all *Paracoccus* strains studied can survive under free-living conditions, it is not surprising that all strains comprise such a diversity of IS elements.

All strains were also screened for genes involved in secondary metabolite biosynthesis using the antiSMASH6.0 database [55]. SM play a key role in mediating competitive or cooperative interactions [56] and can act as weapon against competitors, signaling molecules, agents of symbiosis or as differentiation effectors that could influence niche adaptation [57]. Moreover, all strains were screened for glycine, betaine and trehalose since there are known stress modulators in *Paracoccus* for thermal endurance, salt tolerance, and osmolytic stress [58, 59]. In the six type strains, 50 potential SM clusters were identified (Table 3; S7D Table).

The ectoine cluster, which was found in all strains with 100% similarity, is typically present in bacteria that can endure extreme environmental conditions, such as high salinity, high temperatures, or low water availability [60]. Ectoine serves as a protective molecule that helps organisms combat osmotic stress, UV radiation, and acts as an antioxidant [60, 61]. Besides ectoine, commonly observed solutes including trehalose, betaine-glycine, glycerol, heat shock proteins, and chaperones have been identified as regulators of stress responses under conditions of elevated osmolarity, thermal stress, and desiccation [58, 62, 63]. We summarized all identified genes in S8 Table. All investigated strains can produce and transport trehalose for accumulation by utilizing the conserved TPP/TPS pathway or using the TreY/TreZ pathway except *P. fistulariae* KCTC22803[T] (S8 Table). None is using the TS pathway which was described additionally for *P.* sp. AK26 [59]. In contrast, to production, trehalose transporters (*sugA-sugC*) were found in all strains. Glycogen operon genes (*glgA-glgC*, and *glgX*) have been detected in all genomes except *P. alcaliphilus* DSM8512[T], which, unlike trehalose, is more important for desiccation [63]. It is possible that the higher pH in the alkaline environment affected the stability and functionality of these enzymes for glycogen metabolism, leading to a loss of this pathway. Vice versa, *P. aestuari* DSM 19484[T] isolated from tidal flat sediment in South Korea [10] comprises a higher number of glycogen genes. This habitat is dry twice a day at ebb tide and is accordingly exposed to strong fluctuations (temperature, UV, humidity), where an increased protection against desiccation could be beneficial. In addition, betaine is synthesized via the *betA-betB*, *betI* operon. A variety of transporters for organic osmolytes such as glycine-betaine clusters (*opuAA*, *opuAB*, *opuD*, *proV*, *proX yehW-Z*, and *ousW*) to combat salt and osmolytic stress have been found. Notably, *pro*W and *bet*T transporters were not detected compared to *P.* sp. AK26 [59]. *P. fisulariae* KCTC22803[T] lacks, besides the trehalose pathways also all *yeh*-group glycine-betaine transporter. This could indicate environmental and host-associated genome adaptations. The strain was isolated from the intestine of the bluespotted cornetfish and therefore maybe don't need to produce high amounts of trehalose for accumulation and glycine-betaine because of more stable environmental conditions inside of the host. Another explanation could be that trehalose is provided by the host because transporters are found. Besides, also *P. alcaliphilus* DSM8512[T] revealed an increased number of genes for glycine-betaine and trehalose production which could indicate as already mentioned an adaption to the extreme alkaline environment with higher pH.

Aquaporin was detected in *P. alcaliphilus* DSM8512[T], *P. fisulariae* KCTC22803[T], *P. saliphilus* DSM18447[T], and *P. stylophorae* LMG25392[T] which are described as a regulator for osmotic stress in water movement [59]. *P. seriniphilus* DSM14827[T] comprise no aquaporin indicating that in the specific ecological niche of marine bryozoan *Bugula plumosa* minor water movements are needed. From another bryozoan *Flustra foliaceae* it is known that bacteria colonize the host surface by development of biofilms [64]. Biofilms consist of EPS which could create a physical barrier and reduce water movements. *P. aestuari* DSM 19484[T] with an assumed free-

living lifestyle also don't encode aquaporin, which may be due to the tidal (ebb and flow) habitat, where aquaporins and the lifestyle are not useful.

For thermal endurance genomes were screened for chaperones and heat-shock proteins. As described previously for *Paracoccus*, no Hsp90 heat shock or RNA polymerase sigma E factor which is known as heat stress management factor was found [62]. Instead, chaperones (*hslO*, *grpE*, *groS*, *groL*, *ibpA*, *dnaK* and *dnaJ*), heat shock proteins (*hspQ*, *hsl*R), the transcriptional repressor (*hrcA*), and the RNA polymerase sigma 32 factor (*rpoH*) were identified in all genomes which were also found in some mesophilic *Paracoccus* species [62].

The presence of the ectoine, trehalose, betaine-glycine, glycogen clusters and chaperones in all strains can again be explained by the facultative free-living lifestyle and specific ecological habitats.

At last, to our knowledge, it was not investigated yet, if there is a correlation between bacterial lifestyles and numbers of encoded secondary metabolites. The regulation and production of SM depends on the complexity and dynamics of the environment where bacteria tend to produce a higher diversity and number of SM than bacteria that live in more stable environments [65, 66]. We observed a similar effect in free-living strains, where more clusters to known, substances were identified compared to host-associated strains. In general, we identified similarity to genes encoding for substances that provide fitness advantages, such as carotenoids [67], pyoverdine [68], and bacillibactin [69], as well as compounds that mediate interactions or confer competitive advantages between bacteria and other organisms in their environment, including desotamide [70], bacteriocin [71], sunshinamide [72], lysocin [73], asukamiycins [74], formicamiycins [75], fengycin [76], and chejuenolide [77]. Many of these substances have not been previously described in *Paracoccus* (S7D Table).

Finally, we examined whether MGEs, in terms of genomic flexibility, cluster together in evolutionary hotspots or are evenly distributed throughout the chromosomes. We compared all identified MGEs (excluding IS elements, S7A Table) of the six *Paracoccus* chromosomes (Fig 5).

The chromosome comparison shows that each strain contains a high number of MGEs in addition to the partially high plasmid content (Table 1; up to nine—*P. aestuarii* DSM 19484$^T$). Therefore, we assume that these strains contain a high genomic flexibility, which is one reason for their ability to adapt to different habitats as horizontal gene transfer (HGT) enabled by MGE is the main driver of prokaryotic evolution and adaptation [6]. The regions of differences in all strains were caused by potential phages, genomic islands or SMs and the genes they transfer. Studies with *E. coli* proposed an increasing gradient of integration of prophages along the ori to ter axis [78]. Our data did not support this trend, which may be due to the analysis of six different *Paracocci* belonging to six different species, resulting in a high number of variables, (six different ecological niches, two potential lifestyles) that cannot be accounted for.

## Conclusion

In summary, we increased the number of available closed high-quality *Paracoccus* genomes from 15 to 21 (09.2022). We were able to classify the six type strains carefully phylogenetically within the genus *Paracoccus*. Furthermore, 16 *Paracoccus* spp. were assigned at species level and five misclassifications were uncovered. Pan-genome analysis of six *Paracoccus* genomes representing six ecological niches and two lifestyles revealed an open minimal core which is predominantly encoded by the chromosome. Extra-chromosomal cloud genes displayed a specific functional pattern for each of the strains, but all comprised genes associated to adaptational processes, increasing our understanding of the evolution, ecology, and adaptational potential of the genus *Paracoccus*. With respect to lifestyle adaptations, FL genomes shared

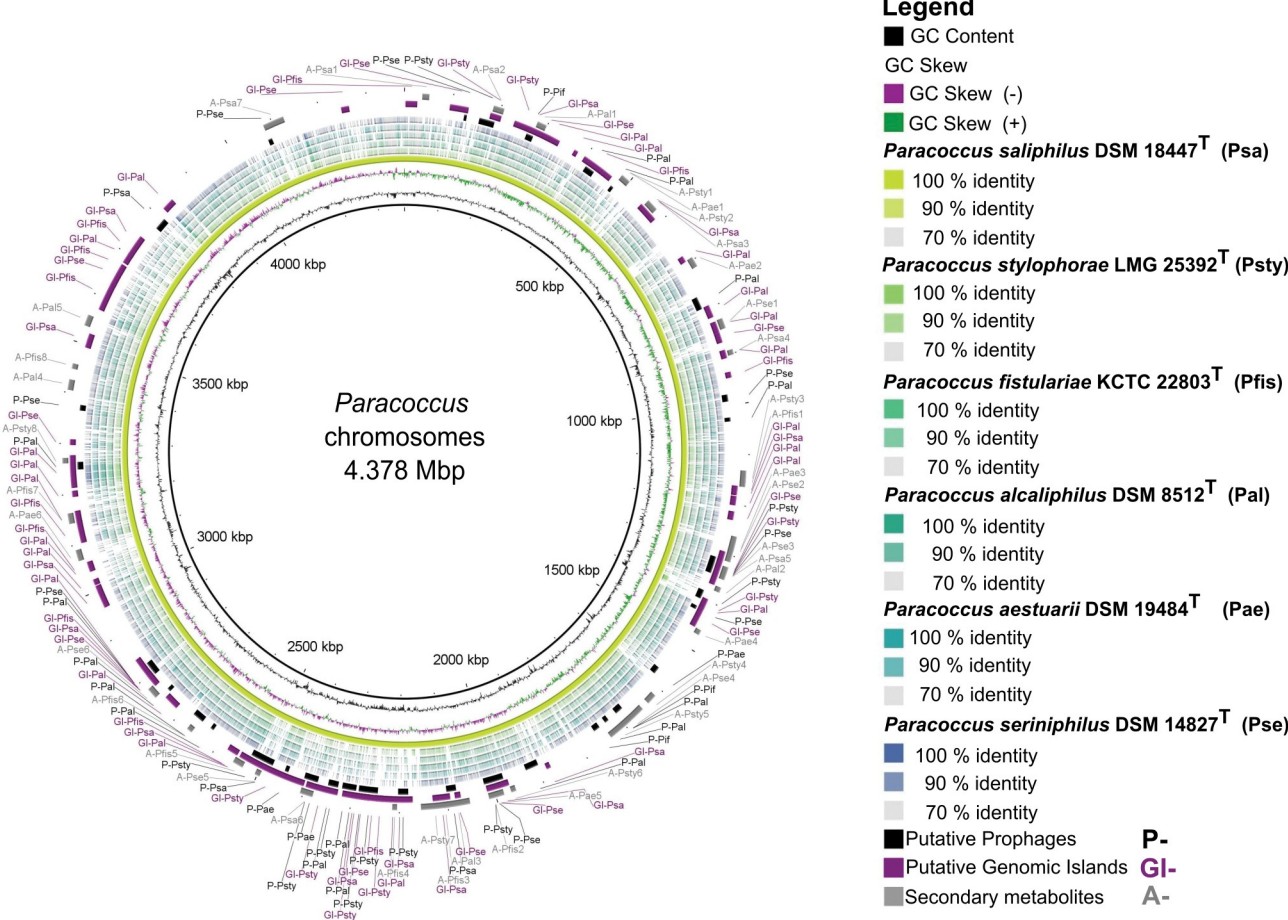

**Fig 5. Comparison of six *Paracoccus* type strain chromosomes.** Concentric colored rings represent Blast matches according to a percentage identity of (100, 90 and 70%). As reference the largest chromosome was chosen (*P. saliphilus* DSM 25392^T) colored in yellow most inner ring. Putative prophage regions of all investigated strains are depicted in black (3 outer ring, P-), putative Genomic Islands in purple (2^nd outer ring, GI-), and secondary metabolite clusters in grey (outer ring, A-). The GC content is shown in black as most inner ring, followed by the GC Skew. The visualization was modified with Inkscape v 1.2. [19].

specific genes involved in genetic exchange via T4SS, which for the genus of *Paracoccus* which are known for comprising a vast majority of extrachromosomal elements, make sense to adapt to the rapid changing environmental conditions. In contrast, in HA genomes genes of diverse function were identified involved in transporting molecules, cell wall modification, attachments, protection against stress, DNA repair and carbon and nitrogen metabolism were identified. Moreover, in-depth comparative genomics of MGEs indicated that free-living *Paracocci* comprised more MGE (GI and IS elements) compared to host-associated strains. In conclusion, complete bacterial type strain genomes are essential as a comprehensive reference for bacterial species and their classification. Pan-genome and comparative MGE analysis guide the understanding of ecological and evolutionary dynamics of bacteria, especially in terms of environmental adaptation. Future studies may benefit from a reduced dataset focusing on a single *Paracoccus* species, such as *P. marcusii* (Fig 2, Cluster 3-*P. marcusii*) with an increased number of strains, to investigate species specific key genes and functional pattern involved in evolution or adaptation effects.

## Material and methods

### Isolation, growth conditions and genomic DNA extraction

In total, six type strain genomes of the genus *Paracoccus* were sequenced in this study all received from type strain collections (S1 Table). All strains except *P. alcaliphilus* DSM 8512[T] were grown in bacto marine broth (MB) medium (DSM 514, https://www.dsmz.de/collection/catalogue/microorganisms/culture-technology/list-of-media-for-microorganisms) for 1–2 days at 20˚C and 100 rpm in the dark. *P. alcaliphilus* DSM 8512[T] was grown under same conditions in *Paracoccus alcaliphilus* medium (DSM 772, https://www.dsmz.de/collection/catalogue/microorganisms/culture-technology/list-of-media-for-microorganisms). Genomic DNA was extracted by using the MasterPure complete DNA and RNA purification kit (Epicentre, Madison, WI) as described by the manufacturer. A pre-lysis step was performed using lysozyme (5 mg, Serva, Heidelberg, Germany) for 30 min at 180 rpm (Infors HT, INFORS AG, Bottmingen, Switzerland) and 37˚C. The quality of the extracted DNA was assessed as described by Hollensteiner et al. 2020 [79].

### Genome sequencing, assembly, and annotation

Illumina paired-end sequencing and Oxford Nanopore sequencing was performed as described by Hollensteiner et al. [79]. Nanopore sequencing was performed for 72 h using a MinION device Mk1B and a SpotON Flow Cell R9.4.1 as recommended by the manufacturer (Oxford Nanopore Technologies) using MinKNOW software (v19.06.8-v20.06.05) for sequencing. For demultiplexing and basecalling Guppy v4.0.15+5694074 (https://community.nanoporetech.com) was applied. Illumina reads were quality-filtered using fastp version 0.20.1 [80] by applying base error correction, removing adapters, read filtering using a quality value (Phred score) >20, with mean quality of 20, and minimum read length of > 50bp. Additionally, a mean quality of 20 was set and all front and ends trimmed below the mean value using a window size of 4. Remaining phiX contaminations were removed with bowtie2 v.2.4.0 [81]. Long reads were quality filtered with fastp version 0.20.1 [80] read filtering using a quality value of 10, and minimum read length of > 1000 bp. Additionally, a mean quality of 10 was set and all front and ends trimmed below the mean value using a window size of 10. Remaining adapters were removed with Porechop v0.2.4 (https://github.com/rrwick/Porechop.git). FastQC v.0.11.8 (https://www.bioinformatics.babraham.ac.uk/projects/fastqc/) was applied for general inspection of read quality. Unicycler version 0.4.8 [82] was used for a *de novo* hybrid assembly in normal mode using the following parameter—no_correct—start_gene_id 70.0—start_gene_cov 70.0. Coverages were calculated for long as well as for short reads using QualiMap v2.2.2 [83]. For mappings minimap2 v2.17 [84] and bowtie2 v.2.4.0 [81] were applied, respectively. For data conversion from.sam to.bam and data sorting SAMtools v.1.9 [85] was used. The overall mean for each genome was calculated. $N_{50}/N_{90}$ for long reads and $N_{50}/L_{90}$ values for the assembly were calculated and summarized in S1 Table. Default parameters were used for all software unless otherwise specified. Genome annotation was performed employing the Prokaryotic Genome Annotation Pipeline (PGAP) v 5.0, accessed 2021-01-04 [86]. BUSCO v5.4.5 was used to evaluate the quality of genomes [17, 87] by using the bacteria_odb10 dataset (OrthoDBv10; 2023-03-03).

### Genome data acquisition and quality control

For an overall taxonomic analysis of the whole genus *Paracoccus* the EzBio list of species was used. The genus comprised over 84 validly published genomes (https://www.ezbiocloud.net/search?tn=paracoccus accessed 2022-09-22). Genomes were downloaded from the National

Center for Biotechnology Information (NCBI) database (accessed 2022-09-22). In total, 185 genomes were downloaded of which 64 were *Paracoccus* type strains or representative genomes. To eliminate redundancy, identical genomes which were independently sequenced by different laboratories with a lower final status (contigs, scaffolds) were excluded (GCA_014656455.1, GCA_003633525.1, GCA_019633685.1, GCA_014164625.1, GCA_000763885.1) (n = 180). Moreover, four of our sequenced strains, namely (*P. aestuarii* DSM 19484[T]; *P alcaliphilus* DSM 8512[T], *P. saliphilus* DSM 18447[T], and *P seriniphilus* DSM 14827[T] were already sequenced but all in draft status and replaced by our high-quality finished genomes (n = 180) (detailed genome comparisons are summarized in S9 Table. Finally, we deposited two new genome sequences of the type strains *P. fistulariae* KCTC 22803[T] and *P. stylophorae* LMG 25392[T] which resulted in a dataset of 182 genomes. Those (n = 182) were further quality checked for completeness and contamination by using checkM v.1.2.0 [88] (S3 Table) and an initial phylogenetic classification was performed using the Genome Taxonomy Database Toolkit (GTDB-Tk v2.1.1) [18] (S2 Table). Statistics of genomes such as number of sequences, size in bp, GC-content (%) were calculated with the perl script gaas_fasta_statistics. pl v1.2.0 from the GAAS toolkit (https://github.com/NBISweden/GAAS). All *Paracoccus* genomes with a completeness >85% and a contamination rate <5% were used for phylogenetic analysis (n = 160). Genomes that did not meet these quality criteria were removed from the analysis (S3 Table, completeness <85% = grey; contamination >5% orange; remaining duplicates = purple).

## Average nucleotide identity analysis

For an initial taxonomic grouping of the six type strains in the genus *Paracoccus* the Type Strain Genome Server (TYGS) [89] was applied based on 16S rRNA level (S1A Fig). Afterwards, a phylogenetic tree of the whole genomes was performed by the comparison of average nucleotide similarity to all available type strain genomes (n = 64, S1B Fig, S10 Table) and all quality filtered *Paracoccus* genomes (n = 160, Fig 2, S4 Table). The average nucleotide identity (ANI) was calculated with pyANI python module v.0.2.11 [90] based on MUMmer [91] using the ANIm mode (S4 Table).

## Pan-genome analysis

The pangenome of the six type strains, and the identification of core and cloud genes were determined by using Roary v3.13.3 [92] with default settings (minimum BLASTP sequence identity of 70% with paralog splitting). To reduce the bias of annotation at date of analysis all genomes were reannotated with the latest version of the Rapid prokaryotic genome annotation pipeline v1.14.6 [93]. Generated protein GFF3 output files produced by prokka were used for downstream analysis using roary [92]. To estimate whether the pangenome was open or closed, we used Heaps' Law: $\eta = \kappa * N^{-\alpha}$ [94], implemented in the R package "micropan" v.2.1 [95], in which $\eta$ is the expected number of genes for a given number of genomes (N), and $\kappa$ and $\alpha$ are constants to fit the specific curve. The exponent $\alpha$ is an indicator for the status of the pangenome open ($\alpha < 1$) or closed ($\alpha > 1$). Additionally, gene presence and absence tables were loaded in R Studio 2022.12.0+353 [96] and pan-genome figures plotted using the script create_pan_genome_plots.R. For functional bar charts the gene presence absence output table (S4 Table) were filtered according to core, shell, and cloud and by genomic localization chromosomal or extra-chromosomal. Proteins were functional annotated with eggNOGmapper v.2.1.9 [97] using default parameter. Visualization was performed in R Studio using "ggplot2"v.3.4.1 [98] and "viridis" v. 0.6.2 [99]. Inkscape v. 1.2.1 was used for modifications [19].

## Comparative genome analysis and detection of mobile genetic elements

To identify genomic islands IslandViewer 4 was employed [100]. Genomic islands were accounted by a size > 8 kb, a different GC content compared to the remaining genome and the detection of common mobile-related elements such as integrases and transposases. To determine insertion sequences and transposases, ISEscan v1.7.2.3 [54] was applied with default parameters. To determine putative prophage regions PHASTER [101] was used. AntiSMASH 6.0 [55] was chosen to detect potential biosynthetic gene clusters encoding for secondary metabolites with relaxed parameters using all extra features. Genome comparison of all six sequenced *Paracoccus* type strains genomes was performed via BRIG [102]. BLAST matches are shown as concentric viridis colored rings on a sliding scale according to percentage identity (100%, 90%, or 70%). Mobile genetic elements such as, genomic islands, phages and potential secondary metabolites were highlighted manually added as tab-separated file.

## Supporting information

**S1 Fig. Phylogenetic tree of investigated six *Paracoccus* type strains.** The trees were reconstructed based on (A) 16S rRNA comparison using TYGS, and (B) WGS comparison with all 64 available *Paracoccus* genomes using ANI. Presented *Paracoccus* genomes used in this study are highlighted in red. Complete genomes are highlighted in blue.
(PDF)

**S1 Table. Sequencing statistics and genomic features of the *Paracoccus* strains.**
(XLSX)

**S2 Table. Quality assessment of *Paracoccus* genomes.**
(XLSX)

**S3 Table. Initial phylogenetic classification of *Paracoccus* genomes using GTDB-TK.**
(XLSX)

**S4 Table. Average nucleotide identity values of *Paracoccus* genomes.**
(XLSX)

**S5 Table. Pan genome analysis of six *Paracoccus* type strains.**
(XLSX)

**S6 Table. Genes associated to lifestyle of *Paracoccus* genomes.**
(XLSX)

**S7 Table. Mobile genetic elements and secondary metabolite gene clusters detected in *Paracoccus* genomes.**
(XLSX)

**S8 Table. Identified genes in *Paracoccus* genomes for thermal endurance, salt tolerance and stress response.**
(XLSX)

**S9 Table. Comparison of six *Paracoccus* type strains to previous sequencing results.**
(XLSX)

**S10 Table. Average nucleotide identity values of all available *Paracoccus* type strain/representative genomes.**
(XLSX)

## Acknowledgments

We thank Janina Leinberger for strain acquisition, raising and shipment. We thank Melanie Heinemann and Mechthild Bömeke for technical support.

## Author Contributions

**Conceptualization:** Thorsten Brinkhoff, Rolf Daniel.

**Data curation:** Jacqueline Hollensteiner.

**Formal analysis:** Jacqueline Hollensteiner, Dominik Schneider, Anja Poehlein, Thorsten Brinkhoff, Rolf Daniel.

**Funding acquisition:** Thorsten Brinkhoff, Rolf Daniel.

**Investigation:** Jacqueline Hollensteiner, Dominik Schneider, Anja Poehlein.

**Project administration:** Thorsten Brinkhoff, Rolf Daniel.

**Supervision:** Rolf Daniel.

**Validation:** Jacqueline Hollensteiner, Dominik Schneider.

**Visualization:** Jacqueline Hollensteiner, Dominik Schneider.

**Writing – original draft:** Jacqueline Hollensteiner.

**Writing – review & editing:** Jacqueline Hollensteiner, Dominik Schneider, Anja Poehlein, Thorsten Brinkhoff, Rolf Daniel.

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
