## [Decision Letter · Decision Letter 0]

15 Aug 2023

PONE-D-23-12094

Pan-genome analysis of six complete Paracoccus type strain genomes from hybrid next generation sequencing

PLOS ONE

Dear Dr. Daniel,

I apologize for the delay in getting back to you with a first decision on your work.

Thank you for submitting your manuscript to PLOS ONE. After careful consideration, we feel that it has merit but does not fully meet PLOS ONE’s publication criteria as it currently stands. Therefore, we invite you to submit a revised version of the manuscript that addresses the points raised during the review process.

As you can see below, both reviewers have highlighted the importance of your study for the field. However, there are several major concerns that will require your attention before the manuscript can be considered further.

The first point is the conclusions the authors have arrived at based on the comparison of only six genomes from type specimens, which was highlighted by both reviewers. Both reviewers provide very helpful suggestions for the authors to move forward (comparison of free-living and host-associated genomes within the same species, inclusion of more genomes for comparison, or the construction a phylogenomic tree using all available 21 genomes and then further selection and analysis of diverse Paracoccus strains based on their phylogenomic positions.

Reviewer 1 further suggested adding a functional enrichment analysis to identify differentially present or absent genes in the COG/KEGG categories, along with further characterization of free-living vs. host-associated features between the species (detailed below in the major comments raised by the reviewer), with which I wholeheartedly agree. Finally, I would like to encourage the authors to pay specific attention to the clarity of their figures, as highlighted by reviewer 2. 

We look forward to receiving your revised manuscript.

Kind regards,

Claudia Isabella Pogoreutz

Academic Editor

PLOS ONE

Journal Requirements:

 Whilst you may use any professional scientific editing service of your choice, PLOS has partnered with both American Journal Experts (AJE) and Editage to provide discounted services to PLOS authors. Both organizations have experience helping authors meet PLOS guidelines and can provide language editing, translation, manuscript formatting, and figure formatting to ensure your manuscript meets our submission guidelines. To take advantage of our partnership with AJE, visit the AJE website (http://aje.com/go/plos) for a 15% discount off AJE services. To take advantage of our partnership with Editage, visit the Editage website (www.editage.com) and enter referral code PLOSEDIT for a 15% discount off Editage services. If the PLOS editorial team finds any language issues in text that either AJE or Editage has edited, the service provider will re-edit the text for free.

Reviewers' comments:

Reviewer's Responses to Questions

**Comments to the Author**

1. Is the manuscript technically sound, and do the data support the conclusions?

Reviewer #1: Partly

Reviewer #2: Partly

2. Has the statistical analysis been performed appropriately and rigorously? 

Reviewer #1: Yes

Reviewer #2: I Don't Know

3. Have the authors made all data underlying the findings in their manuscript fully available?

Reviewer #1: Yes

Reviewer #2: Yes

4. Is the manuscript presented in an intelligible fashion and written in standard English?

Reviewer #1: Yes

Reviewer #2: Yes

5. Review Comments to the Author

Reviewer #1: This study presents a comprehensive analysis of the globally distributed bacteria genus Paracoccus, which occupies diverse ecological niches. The authors conducted phylogenomic analysis to classify previously unclassified Paracoccus strains and identified misclassifications. Additionally, pan-genome analysis was performed on Paracoccus-type strains isolated from various environments, including soils, sediments, and host associations, to investigate the role of lifestyle, adaptation potential, and genomic characteristics. However, there are some conclusions in this study that cannot be directly inferred from the results. Further analysis could be conducted to delve deeper into the comparison of genomes.

Major comments:

1. The authors compared only six complete genomes of different Paracoccus species to examine the differences between free-living and host-associated genomes. However, this result does not convincingly demonstrate the impact of lifestyle or association, as the genomes belong to different species. It is suggested that the authors compare free-living and associated genomes within the same species or include more genomes for comparison.

2. The comparison of COG classification alone does not reveal significant differences of free-living group and associated group. It is recommended that the authors perform functional enrichment analysis on the genomes, specifically identifying differentially present/absent genes in COG and KEGG categories. This analysis can be conducted using anvi’o tool(Eren, A. Murat, et al., Nature microbiology, 2021, ).

3. The authors discuss the genetic differences between free-living and host-associated Paracoccus species. It would be valuable to explore the differences in genes related to cell attachment, chemotaxis, polysaccharide synthesis, and flagella, as well as genes associated with host substance exchange, signaling molecules, and secondary metabolism. The potential functions of Paracoccus-host interaction should also be discussed in the discussion section.

Minor comments:

1. The authors could discuss the relationship between Paracoccus and its hosts, particularly in the context of genome functions.

2. For the re-sequenced four Paracoccus type strains, it would be helpful to compare their genomes with previous sequencing results. Could this information be presented in a table format?

3. In Table 1, consider adding a column to indicate the circularity or strand of contigs.

4. From lines 104-109, the authors could try using the toolkit gtdbtk(Chaumeil, Pierre-Alain, et al. 2020) for classification, as GTDB has an extensive database of genome information. This approach would assign the genomes to the species or genus level.

5. In line 144, please clarify that the reference states the boundary as 95-96%.

6. The order of Figure 4 and Figure 5 appears to be incorrect.

Reviewer #2: Title: Pan-genome analysis of six complete Paracoccus type strain genomes from hybrid next generation sequencing.

Summary: The manuscript by Hollensteiner et al. studied Paracoccus, a globally distributed genus that exhibits high taxonomic, metabolic, and lifestyle diversity, enabling it to thrive in various marine and terrestrial ecological niches. Despite its high taxonomic, metabolic, and lifestyle diversity, little was known about the adaptation strategies of Paracoccus. In this study, the genus was analyzed phylogenomically using 160 strains, resulting in the classification of 16 previously unclassified Paracoccus strains and the identification of five misclassifications. The researchers also performed a pan-genome analysis of Paracoccus strains from different ecological niches, including soils, tidal flat sediment, and host associations. By assembling six complete Paracoccus genomes, the study expanded the collection of high-quality genomes from 15 to 21. They found that Paracoccus genomes possess a high level of genomic plasticity, shaped by various mobile genetic elements, and free-living genomes tend to have larger genomes and more extra-chromosomal elements compared to host-associated strains. The presence of a vast number of adaptive genes enables Paracoccus to rapidly adapt to changing environmental conditions.

General comment: The manuscript by Hollensteiner et al. raises significant concerns regarding the selection of only six Paracoccus type strains for the overall analysis, despite the availability of 21 complete genomes (15 from previous studies and 6 from this study). The authors' rationale for choosing these specific strains based on their various environments and different lifestyles does not appear sufficient, as a pan-core genome analysis of all the complete genomes would provide a better understanding of the Paracoccus genus. Additionally, the exclusion of Paracoccus denitrificans (isolated from soil), the type species of the Paracoccus genus, is questionable since the paper focuses on the pan-genome analysis of type strains. To address these issues, the authors could have either included all 21 complete genomes in their analysis or constructed a phylogenomic tree using the 21 genomes and selected diverse Paracoccus strains based on their phylogenomic positions for further analysis. Alternatively, they could have focused solely on the six Paracoccus strains analyzed without discussing the overall genus status and type strains. Despite these concerns, the computational methods employed in this study were robust. The clarity of certain figures should be enhanced.

Specific comments:

Title (Line numbers 1-2) - Sequencing or assembly method is not crucial in this context; so, remove “from hybrid next generation sequencing” from the title and consider incorporating significant information from the conclusion section, if applicable.

Line numbers 3-4 and 12-13: Is Jacqueline Hollensteiner a corresponding author? The asterisk (*) appears after both Jacqueline Hollensteiner and Rolf Daniel, indicating that they are co-corresponding authors. However, there seems to be an issue as only one corresponding email was provided. Rectify this discrepancy.

Line number 88: Please change the term “finished” to “complete”. In NCBI, the term "complete" is used to refer to a fully sequenced and completely assembled genome.

Line number 89: Change the term “finished” to “complete”.

Line number 98: Change "61.1" to "61.5" to accurately represent the lowest GC content for P. seriniphilus DSM 14827T, which is 61.5.

Line number 110: Under the 'Phylogeny of Paracoccus' section, a phylogenomic tree can be generated using 64 type strains, and the 21 complete genomes can be highlighted within that tree. Alternatively, a separate phylogenomic tree can be constructed based specifically on the 21 complete genomes. Either of these trees would provide valuable insights to readers regarding the diversity of the type strains or complete type strains within the Paracoccus genus.

Line numbers 119-122: Fig 2 legend - These methodologies have already been presented in the “Materials and Methods” section, so remove this part from here.

Line numbers 129-130: Table 2 - “New species classifications” serial number 21 (last row of the Table) - interchange the “Corrected Classification” and “Closest type/representative strain” data. This is a typographical error.

Line number 169: The statement “These results emphasize the highly adaptive genome” - this statement is not appropriate here according to the aforesaid results.

Line number 171: The statement "There is no correlation between lifestyle and cloud gene clusters" is not suitable in its current position. It would be more appropriate to include this statement after presenting the relevant results and findings.

Line number 175: “(Fig3 D)”- correct this typo.

Line number 297: Please search for "glycine betaine" clusters. Glycine betaine is known to assist bacteria in extreme conditions, exhibiting activities such as thermal endurance and salt tolerance. Therefore, conduct a literature search for studies on these activities (thermal endurance and salt tolerance) specifically related to the Paracoccus genus and consider including them in this section.

Line numbers 323-324: Fig 5 legend - These methodologies are already presented in the “Material and Methods” section, so remove this part from here.

Line numbers 360-362: Write the name of the medium as “Paracoccus alcaliphilus medium” and include the complete information as follows: “(DSM 772, https://www.dsmz.de/collection/catalogue/microorganisms/culture-technology/list-of-media for-microorganisms)” in parentheses, similar to the format provided above (Line numbers 357-359).

Line number 374: Remove the URL from the citation as the URL might change in the future, but the paper will remain accessible. Instead, cite the paper as follows: “Shifu Chen, Yanqing Zhou, Yaru Chen, Jia Gu; fastp - an ultra-fast all-in-one FASTQ preprocessor, Bioinformatics, Volume 34, Issue 17, 1 September 2018, Pages i884-i890”. The paper has already been cited in line number 378 as reference number 66.

Line numbers 375-376, 384-385: Please consider using common terms such as "quality value" (Phred score) ≥20 and "minimum length" ≥50 bp, etc. if possible. Employing these simplified terms will facilitate the readers' comprehension of the paper and enhance their study experience.

Line number 392: Format the date “20210104” like the style of line number 393, similar to “2023-03-03”.

Line number 392: The authors have already stated "Default parameters were used for all software unless otherwise specified" in line numbers 389-390. Therefore, remove "with default parameter" from line number 392.

Line number 420: Remove the URL as there is a chance of future alterations, but note that the paper will always be accessible. The authors have already cited a paper for the software (reference number 74).

Line number 455: Write the full form of SRA.

Lines numbers 649-651: Correct the incorrect URL for the cited paper.

Fig 1: “Complete (C) and duplicated” - the abbreviation "(D)" is missing here.

Fig 3: The authors should enhance the quality of this figure as it is currently of low clarity, making it difficult to interpret.

Fig 4: Improve the quality of the figure.

S1 Fig: Increase the font size of the “COG classification” section and the “Number of proteins” on the Y-axis.

S1 Table: Remove the reference URLs and cite the papers using the format “last name et al., year”.

S2 Table: Write the organism's name in the proper format (italics) and superscript the 'T' for type strains. Additionally, add the '%' sign in the completeness, contamination, and strain heterogeneity boxes.

S3 Table: Write the organisms' names in the proper format (italics) and superscript the 'T' for type strains.

S4 Table: Write the organisms' names in the proper format (italics) and include the letter 'T' for type strains.

S5 Table: Superscript the “T” for type strains.

6. PLOS authors have the option to publish the peer review history of their article (what does this mean?). If published, this will include your full peer review and any attached files.

Reviewer #1: **Yes: **Cong Fei

Reviewer #2: No

---

## [Author Response · Author response to Decision Letter 0]

22 Sep 2023

Response to reviewers:

Reviewer #1:

This study presents a comprehensive analysis of the globally distributed bacteria genus Paracoccus, which occupies diverse ecological niches. The authors conducted phylogenomic analysis to classify previously unclassified Paracoccus strains and identified misclassifications. Additionally, pan-genome analysis was performed on Paracoccus-type strains isolated from various environments, including soils, sediments, and host associations, to investigate the role of lifestyle, adaptation potential, and genomic characteristics. However, there are some conclusions in this study that cannot be directly inferred from the results. Further analysis could be conducted to delve deeper into the comparison of genomes.

Major comments:

1. The authors compared only six complete genomes of different Paracoccus species to examine the differences between free-living and host-associated genomes. However, this result does not convincingly demonstrate the impact of lifestyle or association, as the genomes belong to different species. It is suggested that the authors compare free-living and associated genomes within the same species or include more genomes for comparison.

First of all we want to thank the reviewer for the valuable input. 

Response: Of course more genomes would be valuable. However, we here present a valuable starting point how to address this question. For none of our presented species other strains or only one other strain are even available. Based on our analysis we could see that P. marcusii (see Fig 2) is the most promising species for deepen the analysis in the future. 

2. The comparison of COG classification alone does not reveal significant differences of free-living group and associated group. It is recommended that the authors perform functional enrichment analysis on the genomes, specifically identifying differentially present/absent genes in COG and KEGG categories. This analysis can be conducted using anvi’o tool (Eren, A. Murat, et al., Nature microbiology, 2021).

Response: The reviewer is right, and we performed this analysis with anvi’o using the ani-get-enriched-functions-per-pan using COG functions and Kofams. However, it is stated that the group size of three is small and the results needs to be taken with great caution. We inspected the data, and the analysis revealed no q-values below 0.05, which is the border for not ending up with false positives. Therefore, we excluded the data from the manuscript, since the data do not allow robust interpretation. We can support the reviewer with the data if wanted. However, we are not able to use the data for a robust interpretation. Instead, we wrote a section of identified genes specific for each lifestyle (see Table S6, line 199- 243).

3. The authors discuss the genetic differences between free-living and host-associated Paracoccus species. It would be valuable to explore the differences in genes related to cell attachment, chemotaxis, polysaccharide synthesis, and flagella, as well as genes associated with host substance exchange, signaling molecules, and secondary metabolism. The potential functions of Paracoccus-host interaction should also be discussed in the discussion section.

Response: we wrote a new section to address this (see Table S6, (see line 199- 243). We want to highlight, that there is not much literature out for the hosts that have announced for host-associated Paracoccus but did our very best to present and discuss the data adequately. 

Minor comments:

1. The authors could discuss the relationship between Paracoccus and its hosts, particularly in the context of genome functions.

Paracoccus and its host and genome function, 

Response: We wrote a new section to address this (see Table S6, (see line 199- 243). We want to highlight, that there is not much literature out for the hosts that have announced for host-associated Paracoccus but did our very best to present and discuss the data adequately. 

2. For the re-sequenced four Paracoccus type strains, it would be helpful to compare their genomes with previous sequencing results. Could this information be presented in a table format?

Response: We agree that it would be nice to evaluate the differences of previous sequencing results in depth. Therefore, we added to the section of main differences in the manuscript in line 501 to 505, a detailed supplementary table (S9 Table), where all differences with previous sequenced genomes are summarized including: Sequencing Status, Sequencing technology, Number of contigs/scaffold, Genome Size, Genome Coverage, GC, CDS, gene, RNAs, CheckM completeness and CheckM contamination rate. 

3. In Table 1, consider adding a column to indicate the circularity or strand of contigs.

Response: Since all presented genomes are in a complete status, and all contigs regardless of chromosome or plasmid have a circular status, we refer to line 99-101. There we already stated the circularity status for chromosomes but added the status information for all plasmids.

4. From lines 104-109, the authors could try using the toolkit gtdbtk (Chaumeil, Pierre-Alain, et al. 2020) for classification, as GTDB has an extensive database of genome information. This approach would assign the genomes to the species or genus level.

Response: we exactly performed the asked analysis while downloading all genomes for double checking the taxonomy, but we did not include the results. In the past BUSCO instead of GTDB-Tk was more important that is why we implemented that instead. We added the results of the GTDB-Tk analysis and rewrote the section (see line 105-107; line 111-115, and S2 Table). We also added a section in the Material and Method section (see line 506-509 with corresponding reference).

5. In line 144, please clarify that the reference states the boundary as 95-96%.

Response: We rephrased the sentence (see line 147-149) to emphasize that the species boundary is reflected by 94-96 %. This includes the boundary proposed by Richter et al. 2009 (95-96%) but also includes the ANI value of ~94% which corresponds to the traditional 70% DNA-DNA standard proposed Konstantinidis et al. 2005. 

6. The order of Figure 4 and Figure 5 appears to be incorrect.

Response: The reviewer is right, we corrected it accordingly.

Reviewer #2: 

Title: Pan-genome analysis of six complete Paracoccus type strain genomes from hybrid next generation sequencing.

Summary: The manuscript by Hollensteiner et al. studied Paracoccus, a globally distributed genus that exhibits high taxonomic, metabolic, and lifestyle diversity, enabling it to thrive in various marine and terrestrial ecological niches. Despite its high taxonomic, metabolic, and lifestyle diversity, little was known about the adaptation strategies of Paracoccus. In this study, the genus was analyzed phylogenomically using 160 strains, resulting in the classification of 16 previously unclassified Paracoccus strains and the identification of five misclassifications. The researchers also performed a pan-genome analysis of Paracoccus strains from different ecological niches, including soils, tidal flat sediment, and host associations. By assembling six complete Paracoccus genomes, the study expanded the collection of high-quality genomes from 15 to 21. They found that Paracoccus genomes possess a high level of genomic plasticity, shaped by various mobile genetic elements, and free-living genomes tend to have larger genomes and more extra-chromosomal elements compared to host-associated strains. The presence of a vast number of adaptive genes enables Paracoccus to rapidly adapt to changing environmental conditions.

General comment: The manuscript by Hollensteiner et al. raises significant concerns regarding the selection of only six Paracoccus type strains for the overall analysis, despite the availability of 21 complete genomes (15 from previous studies and 6 from this study). The authors' rationale for choosing these specific strains based on their various environments and different lifestyles does not appear sufficient, as a pan-core genome analysis of all the complete genomes would provide a better understanding of the Paracoccus genus. Additionally, the exclusion of Paracoccus denitrificans (isolated from soil), the type species of the Paracoccus genus, is questionable since the paper focuses on the pan-genome analysis of type strains. To address these issues, the authors could have either included all 21 complete genomes in their analysis or constructed a phylogenomic tree using the 21 genomes and selected diverse Paracoccus strains based on their phylogenomic positions for further analysis. Alternatively, they could have focused solely on the six Paracoccus strains analyzed without discussing the overall genus status and type strains. Despite these concerns, the computational methods employed in this study were robust. The clarity of certain figures should be enhanced.

First of all, we want to thank the reviewer for the valuable input. 

Response: We understand the reviewer's concerns about strain selection, but at the beginning of the analysis we have tried to establish a starting point with a manageable set of genomes to build a robust computational workflow (which the reviewer kindly acknowledges) and, of course, to accompany the data analysis of the entire genus Paracoccus was performed by Puri et al. 2020, but we clearly saw problems in implementing such a substantial number for establishing a workflow where most strains are not validly described and simply lack important metadata such as clear lifestyle assignments or association. Other considerations were the high “background noise” of the described open core at the genus level itself. Therefore, we did at the beginning an analysis on 16SrRNA (please see newly provided Fig 1A) and picked three strains by respecting lifestyles, from the two biggest deep branching Paracoccus groups. We tried to pick diverse as possible by generating and investigating novel data by using same consistent sequencing techniques (to minimize or keep biases consistent).

However, during our analysis we recorded a very promising species in our data, namely P. marcusii (see Figure 2) which is a biogeographically highly distributed species in various ecological niches. We already planned and started to answer exactly this question at species level to exclude unimportant background noise from genus level. However, this would expand this publication is beyond what we can present here. Nevertheless, we added the questioned phylogenetic tree of 66 Paracoccus genomes in which the 21 genomes are highlighted as requested but would stick to our analysis of the presented six Paracoccus genomes by the stated reasons. We respected and presented all available Paracoccus genomes just to carefully classify our strains in the Paracoccus group because it is simply important to do so. As we could show, in the past many studies lack a classification or are misclassified. 

Specific comments:

Title (Line numbers 1-2) - Sequencing or assembly method is not crucial in this context; so, remove “from hybrid next generation sequencing” from the title and consider incorporating significant information from the conclusion section, if applicable.

Response: We adjusted the title to Pan-genome analysis of six Paracoccus genomes reveal lifestyle traits 

Line numbers 3-4 and 12-13: Is Jacqueline Hollensteiner a corresponding author? The asterisk (*) appears after both Jacqueline Hollensteiner and Rolf Daniel, indicating that they are co-corresponding authors. However, there seems to be an issue as only one corresponding email was provided. Rectify this discrepancy.

Response: During upload the decision was made that only one corresponding author is sufficient. We removed the asterisk after Jacqueline Hollensteiner (line 3).

Line number 88: Please change the term “finished” to “complete”. In NCBI, the term "complete" is used to refer to a fully sequenced and completely assembled genome.

Was corrected in the manuscript.

Line number 89: Change the term “finished” to “complete”.

Was corrected in the manuscript.

Line number 98: Change "61.1" to "61.5" to accurately represent the lowest GC content for P. seriniphilus DSM 14827T, which is 61.5.

Response: The GC was corrected (line 103) as recommended.

Line number 110: Under the 'Phylogeny of Paracoccus' section, a phylogenomic tree can be generated using 64 type strains, and the 21 complete genomes can be highlighted within that tree. Alternatively, a separate phylogenomic tree can be constructed based specifically on the 21 complete genomes. Either of these trees would provide valuable insights to readers regarding the diversity of the type strains or complete type strains within the Paracoccus genus.

Response: We agree to the reviewer suggestions and already prepared those phylogenetic trees at initial analysis but excluded it from the presented data. Since we highlight in our manuscript “reclassifications or wrongly assigned taxa” we would like to keep the Fig2 as it is. 

But we prepared an additional Figure “S1 Fig” addressing all 64 type strains by adding two new genomes (and replacing redundant type strains (n=66). Moreover, we also did an initial taxonomic classification based on 16S rRNA using the Type Strain Genome Server (TYGS) (https://tygs.dsmz.de/) which is visualized in S1 FigA. The phylogenetic visualization on WGS is depicted in S1 FigB. 

We highlighted our genomes as well as all complete genomes as requested and added a section in the material and method section (line 517-523).

Line numbers 119-122: Fig 2 legend - These methodologies have already been presented in the “Materials and Methods” section, so remove this part from here.

The method part was streaken out of the Fig 2 legend (see line 123-126).

Line numbers 129-130: Table 2 - “New species classifications” serial number 21 (last row of the Table) - interchange the “Corrected Classification” and “Closest type/representative strain” data. This is a typographical error.

Response: Thank you for recognizing. We corrected the typographical error.

Line number 169: The statement “These results emphasize the highly adaptive genome” - this statement is not appropriate here according to the aforesaid results.

Response: We do not agree that our statement is not appropriate here. Based on pangenome analysis, a high cloud genome (76,95 %) and a small core (8,84 %) is an indicator for highly adaptive genomes. In pangenome analysis, the "core genome" consists of genes shared by all strains within a species, while the "cloud genome" includes genes present in only a subset of strains. A high cloud genome indicates a diverse set of genes that are not present in all strains, suggesting genetic variability and genomic diversity within the population. A small core genome implies that there are fewer genes shared universally, which may suggest the presence of specialized genes that contribute to adaptability to different environments or conditions. Therefore, a combination of a high cloud genome and a small core genome can indeed be associated with genomes that have a greater potential for adaptation to varying conditions. 

However, we double checked if such statements were made before based on such analysis and found that Yisong Li et al. 2021 [1], Chibani C. et al. 2020 [2], Dziewit et al. 2014 [3], and Tantoso E. et al. 2022 [4] also made such statements based on their data.

Line number 171: The statement "There is no correlation between lifestyle and cloud gene clusters" is not suitable in its current position. It would be more appropriate to include this statement after presenting the relevant results and findings.

Response: The reviewer is right. We here again completely restructured the manuscript to implement all requested analysis to provide a more robust statement based on the presented data. (see line 199-243, Table S6). 

Line number 175: “(Fig3 D)”- correct this typo.

Response: We corrected the typo.

Line number 297: Please search for "glycine betaine" clusters. Glycine betaine is known to assist bacteria in extreme conditions, exhibiting activities such as thermal endurance and salt tolerance. Therefore, conduct a literature search for studies on these activities (thermal endurance and salt tolerance) specifically related to the Paracoccus genus and consider including them in this section.

Response: We agree to the reviewer and added a complete new section (see line 341-343, line 348-393; S8 Table). References were added, respectively. 

Line numbers 323-324: Fig 5 legend - These methodologies are already presented in the “Material and Methods” section, so remove this part from here.

Response: We removed the redundant method section form the figure capture.

Line numbers 360-362: Write the name of the medium as “Paracoccus alcaliphilus medium” and include the complete information as follows: “(DSM 772, https://www.dsmz.de/collection/catalogue/microorganisms/culture-technology/list-of-media for-microorganisms)” in parentheses, similar to the format provided above (Line numbers 357-359).

Response: We corrected the section as suggested (line 457-459).

Line number 374: Remove the URL from the citation as the URL might change in the future, but the paper will remain accessible. Instead, cite the paper as follows: “Shifu Chen, Yanqing Zhou, Yaru Chen, Jia Gu; fastp - an ultra-fast all-in-one FASTQ preprocessor, Bioinformatics, Volume 34, Issue 17, 1 September 2018, Pages i884-i890”. The paper has already been cited in line number 378 as reference number 66.

Response: We removed the URL and added the reference.

Line numbers 375-376, 384-385: Please consider using common terms such as "quality value" (Phred score) ≥20 and "minimum length" ≥50 bp, etc. if possible. Employing these simplified terms will facilitate the readers' comprehension of the paper and enhance their study experience.

Response: We understand the argumentation of the reviewer and rewrote the section (line 470-474). We also rephrased section for long read trimming (line 475-478).

Line number 392: Format the date “20210104” like the style of line number 393, similar to “2023-03-03”.

Response: corrected.

Line number 392: The authors have already stated "Default parameters were used for all software unless otherwise specified" in line numbers 389-390. Therefore, remove "with default parameter" from line number 392.

Response: corrected.

Line number 420: Remove the URL as there is a chance of future alterations, but note that the paper will always be accessible. The authors have already cited a paper for the software (reference number 74).

Response: the URL was removed.

Line number 455: Write the full form of SRA.

Response: SRA was changed into Sequence Read Archive

Lines numbers 649-651: Correct the incorrect URL for the cited paper.

Response: We corrected and adjusted all references using the PLOSOne style.

Fig 1: “Complete (C) and duplicated” - the abbreviation "(D)" is missing here.

Response: We corrected the figure.

Fig 3: The authors should enhance the quality of this figure as it is currently of low clarity, making it difficult to interpret.

Response: We will double check. We prepared all figures in high resolution 1200dpi.svgs and need to reduce it to 600dpi.tiff formats while data uploading. However, also those figures had still a good resolution and no hints for low clarity. 

Fig 4: Improve the quality of the figure.

Response: We will double check. We prepared all figures in high resolution 1200dpi.svgs and need to reduce it to 600dpi.tiff formats while data uploading. However, also those figures had still a good resolution and no hints for low clarity. 

S1 Fig: Increase the font size of the “COG classification” section and the “Number of proteins” on the Y-axis.

Response: We increased the font size as the reviewer stated.

S1 Table: Remove the reference URLs and cite the papers using the format “last name et al., year”.

Response: The URLs were removed, and the papers cited as desired.

S2 Table: Write the organism's name in the proper format (italics) and superscript the 'T' for type strains. Additionally, add the '%' sign in the completeness, contamination, and strain heterogeneity boxes.

Response: Table S2 (now Table S3) was corrected as suggested.

S3 Table: Write the organisms' names in the proper format (italics) and superscript the 'T' for type strains.

Response.Table S3 (now Table S4) was corrected as suggested.

S4 Table: Write the organisms' names in the proper format (italics) and include the letter 'T' for type strains.

Response: Was corrected (now S5 Table)

S5 Table: Superscript the “T” for type strains.

Response: corrected (now S6 Table).

---

## [Decision Letter · Decision Letter 1]

16 Nov 2023

Pan-genome analysis of six Paracoccus type strain genomes reveal lifestyle traits

PONE-D-23-12094R1

Dear Dr. Daniel,

We’re pleased to inform you that your manuscript has been judged scientifically suitable for publication and will be formally accepted for publication once it meets all outstanding technical requirements.

Kind regards,

Gabriel Moreno-Hagelsieb

Academic Editor

PLOS ONE

Additional Editor Comments (optional):

The reviewer has noted that you addressed all of their comments. I agree. It is still be a very good idea to check the manuscript for missing commas and periods, since I found several sentences that ran too long without any punctuation.

Reviewers' comments:

Reviewer's Responses to Questions

**Comments to the Author**

1. If the authors have adequately addressed your comments raised in a previous round of review and you feel that this manuscript is now acceptable for publication, you may indicate that here to bypass the “Comments to the Author” section, enter your conflict of interest statement in the “Confidential to Editor” section, and submit your "Accept" recommendation.

Reviewer #2: All comments have been addressed

2. Is the manuscript technically sound, and do the data support the conclusions?

Reviewer #2: Partly

3. Has the statistical analysis been performed appropriately and rigorously? 

Reviewer #2: Yes

4. Have the authors made all data underlying the findings in their manuscript fully available?

Reviewer #2: Yes

5. Is the manuscript presented in an intelligible fashion and written in standard English?

Reviewer #2: Yes

6. Review Comments to the Author

Reviewer #2: (No Response)

7. PLOS authors have the option to publish the peer review history of their article (what does this mean?). If published, this will include your full peer review and any attached files.

Reviewer #2: No

---

## [Editor Report · Acceptance letter]

8 Dec 2023

PONE-D-23-12094R1 

Pan-genome analysis of six *Paracoccus* type strain genomes reveal lifestyle traits 

Dear Dr. Daniel:

I'm pleased to inform you that your manuscript has been deemed suitable for publication in PLOS ONE. Congratulations! Your manuscript is now with our production department. 

Kind regards, 

on behalf of

Prof. Gabriel Moreno-Hagelsieb 

Academic Editor

PLOS ONE